# Higher-order connectomics of human brain function reveals local topological signatures of task decoding, individual identification, and behavior

Andrea Santoro [1,2] ✉, Federico Battiston [3], Maxime Lucas [2,4], Giovanni Petri [2,5,6] & Enrico Amico [1,7,8] ✉

Traditional models of human brain activity often represent it as a network of pairwise interactions between brain regions. Going beyond this limitation, recent approaches have been proposed to infer higher-order interactions from temporal brain signals involving three or more regions. However, to this day it remains unclear whether methods based on inferred higher-order interactions outperform traditional pairwise ones for the analysis of fMRI data. To address this question, we conducted a comprehensive analysis using fMRI time series of 100 unrelated subjects from the Human Connectome Project. We show that higher-order approaches greatly enhance our ability to decode dynamically between various tasks, to improve the individual identification of unimodal and transmodal functional subsystems, and to strengthen significantly the associations between brain activity and behavior. Overall, our approach sheds new light on the higher-order organization of fMRI time series, improving the characterization of dynamic group dependencies in rest and tasks, and revealing a vast space of unexplored structures within human functional brain data, which may remain hidden when using traditional pairwise approaches.

Since the introduction of the concept of connectome in neuroscience[1,2], network models have become fundamental tools to study human brain structure and function[3,4]. In these formal models[5–7], nodes represent neurons, brain regions, or sensors, and edges represent relationships between them—e.g., a statistical relationship, or physical connectivity. Functional connectivity (FC) is arguably the most widely used type of brain network model[8–10]: in whole brain modeling, it defines (weighted) edges as statistical dependencies between time series recordings associated with brain regions, such as those obtained using functional magnetic resonance imaging (fMRI).

Typically, the FC network is then analyzed through a plethora of metrics offered by network theory[11–13]. Despite its widespread adoption, FC is—like traditional network models—limited by its underlying hypothesis that interactions between nodes are strictly pairwise. In other words, it assumes that the brain can be described solely by pairwise relationships between its neurons or regions. However, higher-order interactions (HOIs)[14–16]—relationships that involve three or more nodes simultaneously—are important to fully characterize the complex spatiotemporal dynamics of the human brain, as suggested by mounting evidence at both micro- and macro-scales[17–26]. Indeed,

[1]Neuro-X Institute, EPFL, Geneva, Switzerland. [2]CENTAI, Turin, Italy. [3]Department of Network and Data Science, Central European University, Vienna, Austria. [4]Department of Mathematics & Namur Institute for Complex Systems (naXys), Université de Namur, Namur, Belgium. [5]NPLab, Network Science Institute, Northeastern University London, London, UK. [6]Department of Physics, Northeastern University, Boston, MA, USA. [7]School of Mathematics, University of Birmingham, Birmingham, UK. [8]Centre for Human Brain Health, University of Birmingham, Birmingham, UK. ✉e-mail: andrea.santoro@centai.eu; e.amico@bham.ac.uk

even in simple dynamical systems, recent theoretical findings indicate that the presence of higher-order interactions[27,28], typically represented by network generalizations such as simplicial complexes or hypergraphs[14,29], can exert profound qualitative shifts in a system's dynamics[30–34]. As such, methods relying on pairwise statistics alone might be insufficient, as significant information might only be present or detectable in the joint probability distributions and not in the pairwise marginals, therefore failing to identify higher-order behaviors[28].

Recent advancements in technology have allowed us to record higher-order phenomena at the micro-scale neuronal level, such as the simultaneous firing of groups of neurons in mice[35–37] or monkeys[38]. However, this type of data is not yet available in humans for studying large-scale brain dynamics. Instead, we typically rely on non-invasive techniques like M/EEG and fMRI, which provide only noisy estimates of the neural activity and cannot directly capture "true" higher-order brain functions. Thus, researchers must rely on statistical methods to infer higher-order interactions from neuroimaging signals recorded from regions of interest. Recent approaches, whether rooted in information theory[39,40] or computational topology[41–44], have only just begun to provide evidence that these HOIs exist in the brain, that they can be reconstructed from fMRI, and that they can significantly contribute to explaining the complex dynamics of brain function. These emerging approaches, applicable to both healthy[45] and clinical populations[46,47], therefore represent a fundamental shift from methods like motif analysis or classical approaches like Independent Component Analysis (ICA)[48,49]. Initial applications of information-theoretic techniques in fMRI suggested that higher-order dependencies reconstructed from fMRI data can encode meaningful brain bio-markers, including the ability to differentiate patients in different states of consciousness[20,23] or detect effects associated with age[50]. Furthermore, early examples of inferred temporal HOI statistics[41] have been successfully used as features in machine learning classifiers to detect financial crises and classify disease types based on their spatial spreading patterns, providing better accuracy when compared to measures based on pairwise descriptions (lower order), such as edge-

centric approaches[51,52]. Currently, edge-centric approaches in fMRI have shown promise in identifying overlapping brain communities and estimating dynamic connectivity at finer timescales[53] compared to classical FC methods. In contrast, the potential benefits of HOIs in fMRI data analysis have received limited exploration. Specifically, it remains uncertain whether reconstructed HOIs offer advantages and deeper insights over conventional methods for fMRI data analysis.

In this study, we address this question by leveraging a recent topological approach capable of reconstructing HOI structures at the temporal level[41]. Our analysis focuses on resting state and tasks fMRI data from 100 unrelated subjects of the Human Connectome Project (HCP). We show here that *local* higher-order indicators, extracted from instantaneous topological descriptions of the data, outperform traditional node and edge-based methods[51,52] in task decoding as well as providing improved functional brain fingerprinting[54,55] based on local topological structures, and a more robust association between brain activity and behavior. Interestingly, we also find that similar indicators, when defined at the *global* scale, do not significantly outperform traditional pairwise methods, suggesting a localized and spatially-specific role of higher-order functional brain coordination with respect to pairwise connectivity.

## Results

We rely on a recent topological method[41] that combines topological data analysis[56,57] and time series analysis to reveal instantaneous higher-order patterns in fMRI data. This approach builds upon prior work involving edge-level signals and the extension of functional connectivity research beyond pairs[51,52], leveraging low-order signals to define higher-order ones (a reconstruction task that is in general an ill-posed inverse problem). The approach consists of four key steps, as depicted in Fig. 1a–d. (i) We start by standardizing the $N$ original fMRI signals through $z$-scoring (Fig. 1a). (ii) We then compute all possible $k$-order time series as the element-wise products of $k+1$ of these $z$-scored time series, which are further $z$-scored for cross-$k$-order comparability (Fig. 1b). These $k$-order time series represent the instantaneous co-fluctuation magnitude of the associated $(k+1)$-node

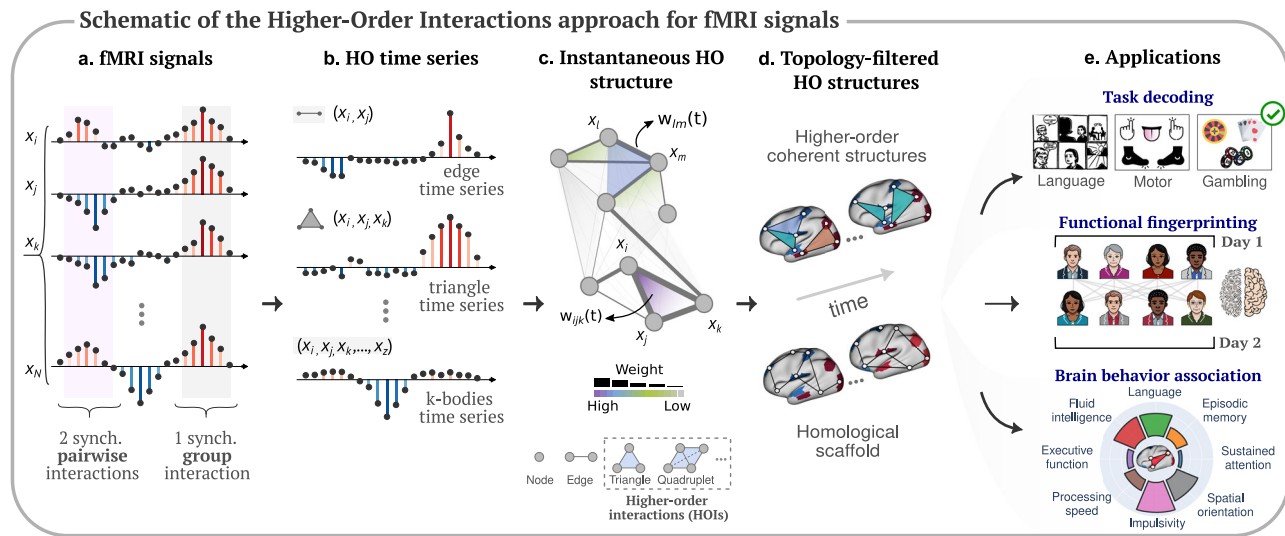

**Fig. 1 | Higher-order brain mapping: schematic of the approach and applications. a** We start from fMRI brain data, which can be encoded into brain signals (i.e., Blood Oxygen Level Dependent (BOLD) fluctuations) from $N$ regions. Signals can encode synchronized pairwise interactions (highlighted in light purple) and group interactions (highlighted in gray). **b** After reconstructing the co-fluctuation time series at each group order (i.e., edges, triangles, $k$-bodies, etc), **c** our instantaneous topological framework enables a time-resolved exploration of the significance of group interactions in comparison to pairwise interactions, encoded

within a simplicial structure, using topological data analysis tools. **d** From this structure, we then extract instantaneous snapshots of brain dynamics in terms of two different topological structures, namely, the higher-order coherent structures and the homological scaffold. **e** To gain more insights into the importance of these higher-order structures during rests or cognitive tasks, we consider three different neuroscience applications, such as: fMRI task decoding, functional brain fingerprinting, and brain-behavior association.

interactions, such as edges and triangles. We finally assign a sign to the resulting $k$-order time series at each time based on a strict parity rule: positive for fully concordant group interactions (nodes times series have all positive or all negative values at that timepoint), and negative for discordant interactions (a mixture of positive and negative values). Notice that this sign remapping allows us to explicitly focus on perfectly coherent contributions, which are always marked as positive (see also Methods for the analytical formulation). *(iii)* For each time $t$, we encode all instantaneous $k$-order (co-fluctuation) time series into a single mathematical object: a weighted simplicial complex (Fig. 1c). We define the weight of each simplex as the value of the associated $k$-order time series at that timepoint. *(iv)* Finally, at each time $t$, we apply computational topology tools to analyze the weights of the simplicial complex and extract two global and two local indicators (see Methods). The rationale for this is that we are interested in assessing the capacity of higher-order indicators to encode human brain function along two axes: the first one, along the spatial gradient (i.e., whole-brain versus local connectivity structures); the second, along the complexity gradient (i.e., low- versus higher-order functional interactions).

## Global and local higher-order task encoding

With respect to the first axis, we consider two global higher-order indicators. The first global indicator, *hyper-coherence*, quantifies the fraction of higher-order triplets that co-fluctuate more than what is expected from the corresponding pairwise co-fluctuations. They are identified in the filtration process as "violating triangles", whose standardized simplicial weight is greater than those of the corresponding pairwise edges. As a second global indicator, we consider the different contributions to the topological complexity coming from the landscape of coherent and decoherent signals, distinguishing between coherent and incoherent co-fluctuations across three contributions (Fully Coherent (FC), Coherent Transition (CT), and Fully Decoherent (FD)) (see "Methods" for a detailed description).

For local indicators, we use the list and weights of violating triangles $\Delta_v$ and the homological scaffolds, both obtained during step *(iv)* of the topological pipeline (see Methods for the formal definitions of these measures). Intuitively, the *identity and weights of the violating triangles* provide insights into higher-order coherent co-fluctuations that cannot be described in terms of pairwise connections (i.e., edges). The *homological scaffold*[18] instead assesses the relevance of edges toward mesoscopic topological structures within the higher-order co-fluctuation landscape. In simpler terms, it is a weighted graph that highlights the importance of certain connections in the overall brain activity patterns, particularly when considering topological structures like 1-dimensional cycles (see Methods for a formal definition). To assess the level of complexity of brain dynamics along the second axis (lower- versus higher-order interactions), we also evaluate two classical and lower-order instantaneous measures: the $N$ (where $N$ is the number of brain nodes) Blood-Oxygen-Level-Dependent (BOLD) time series, and the corresponding edge time series derived from it[51] (see "Methods"). In our analyses, we consider a cortical parcellation of 100 cortical[58] and 19 sub-cortical brain regions as provided by the HCP release[59], for a total of $N = 119$ regions of interest.

For each of the four local methods, which we refer to as BOLD, edges, triangles, and scaffold signals, we construct recurrence plots. To do so, we concatenate the first 300 volumes of resting-state fMRI data with the data from the seven fMRI tasks, excluding the rest blocks, thereby creating a unified fMRI recording, over which we can then compute time-time correlation matrices for the local indicators. The resulting correlation matrices, commonly referred to as recurrence plots in dynamical systems[60], encode in the entry $(i, j)$ Pearson's correlation[61] between the temporal activation at two distinct time points $t_i$ and $t_j$ for a given local indicator. We further process these

matrices by binarizing them at the 95th percentile of their respective distributions, i.e., setting the entries lower than the threshold equal to zero and the remaining elements as one, and apply the Louvain algorithm[62] to identify communities (results for other two thresholds are reported in SI Figs. S1, 2). To evaluate how effectively these community partitions identify the timings corresponding to task and rest blocks, we use the element-centric similarity (ECS) measure[63]. The ECS is a measure of similarity between two community partitions: 0 indicates bad task decoding (entirely dissimilar partitions), while 1 indicates perfect task identification (identical partitions).

In Fig. 2 we show the results of computing both global and local higher-order topological indicators on a resting-state and task-related fMRI dataset containing 100 unrelated subjects from the HCP[64,65] (see details of the HCP dataset in "Methods"). We find that the global whole-brain measures exhibit remarkably similar values across tasks (Fig. 2a, b), with no significant differences ($p$-values exceed 0.1 for all pairwise t-tests), indicating that global topological indicators are not suitable as effective features for task decoding. At the local level, instead, we find that both the violating triangles and the homological scaffolds recurrence plots effectively decode between individual tasks and rest blocks, unlike low-order methods (Fig. 2c–f). More interestingly, we observe that task differentiability increases as we transition from lower-order methods (e.g., edge and BOLD activity) to higher-order ones (e.g., violating triangles and scaffold activity), as evidenced by the increase in the ECS values (Fig. 2c–f). In other words, there is a high degree of discrimination between tasks viewed through the lens of higher-order approaches. This temporal decoding is progressively lost as we move towards BOLD activity. Thus, *local higher-order indicators* provide enhanced quantitative means to understand the temporal evolution of task-related brain activity, underscoring their efficacy in decoding tasks more effectively.

## Functional brain fingerprinting

We now turn our attention to a second application: functional brain fingerprinting[54,55,66]. That is, the ability to identify a subject from a group, solely based on their brain's distinctive functional pattern.

Similarly to the previous section, we aim to compare the four different methods on the HCP dataset relying on two sessions of resting-state fMRI data (i.e., sessions *REST_1_LR* and *REST_2_LR* for test and retest, respectively). We assess the quality of fingerprinting using the differential identifiability measure[55]. This measure quantifies the similarity of brain patterns of individuals with themselves and with others, between the two sessions, averaged over all subjects. Higher values of the measure indicate higher differentiability of the subjects (see Methods for a formal definition). For the temporal measures extracted through the four different methods, namely, BOLD, edges, triangles, and scaffold, we compute static measures of connectivity, resulting in node FC[1], edge functional connectivity (eFC)[52], average violating triangles (triangles), and average scaffold (scaffold). For each method, we rely on such connectivity measures to compute the brain's functional similarity between and within subjects over the two resting-state fMRI sessions to evaluate the differential identifiability score (refer to "Methods" for additional details).

Similarly to the previous analysis, we first compare the four methods focusing on whole-brain connections. In this scenario, the identifiability scores of the four methods exhibit no significant differences, with all methods yielding almost identical results (see SI Fig. S3). On the contrary, when repeating the analysis considering only specific local connections, the methods' performances start to differ. Specifically, for each method, we consider only the connections that involve at least one node within one of the seven resting-state functional networks[8] and subsequently compute the identifiability score. Notably, as reported in Fig. 3a, the triangle method consistently outperforms the other methods for nearly all the seven functional networks

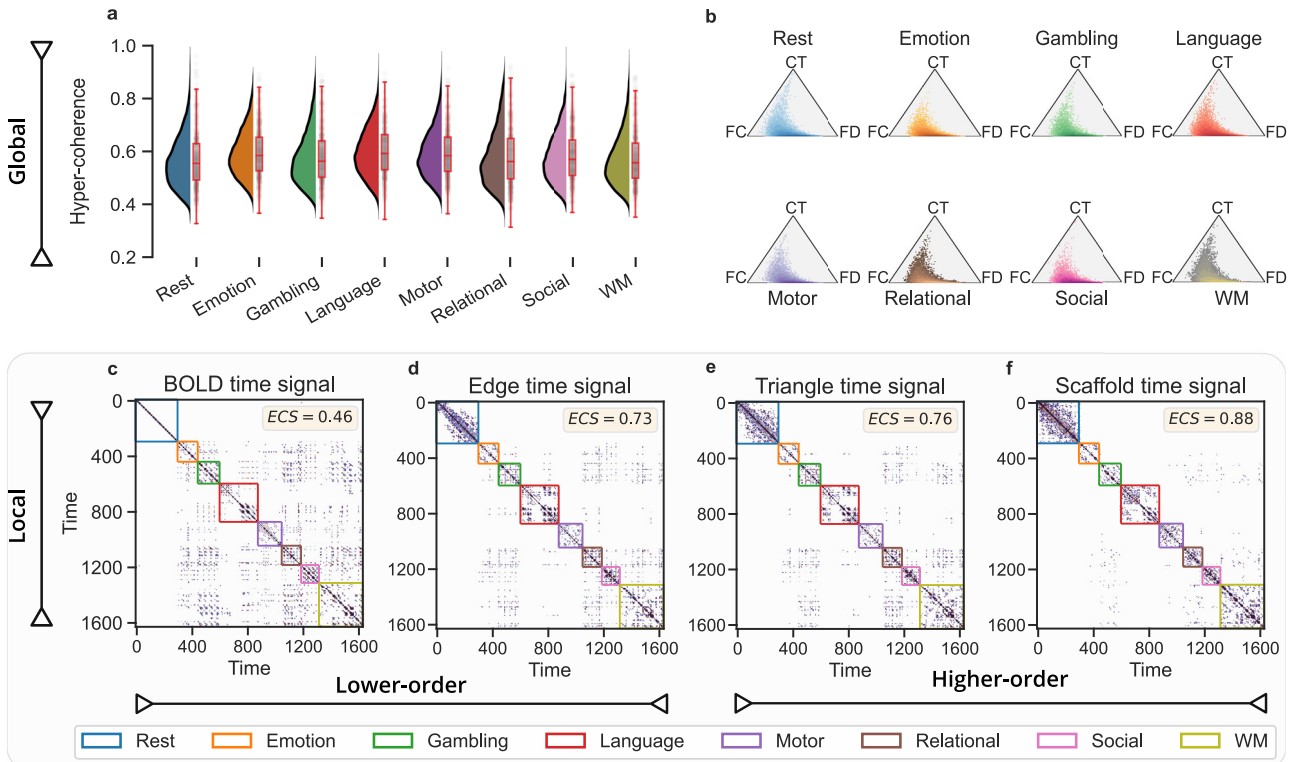

**Fig. 2 | Global and local higher-order topological indicators for fMRI task differentiability. a** We report the distribution of the global hyper-coherence for the concatenated fMRI data at rest and the seven HCP tasks, namely: emotion, gambling, language, motor, relational, social, and working memory (WM). Distributions are obtained when considering the global scores for the 100 unrelated subjects. **b** Two-dimensional histogram of the global hyper-complexity contributions associated with 1D cycles in the landscape of coherent and decoherent co-fluctuations for rest and task fMRI data. Here, the position of each point in the triangle is determined by the three different contributions associated with the 1D cycles, namely, full coherence (FC), coherence transition (CT), and full decoherence (FD) contributions. Remarkably, the two global measures are not able to decode between the different tasks. **c**–**f** We then compare *local* observables by comparing the temporal recurrence plots (i.e., time-time correlation matrices) for the four methods, from the lower-order methods (BOLD and edge signals) to higher-order

ones (triangle and scaffold time signals). We set a common threshold at the 95th percentile to binarize the data when analyzing an fMRI temporal signal obtained by concatenating resting-state and seven HCP tasks. Colored boxes within the plots denote the ground truth of rest and task segments. When comparing the communities identified with the Louvain algorithm[62] against the ground truth partition, we observed that higher-order methods are able to temporally discriminate the different task blocks better than lower-order ones, as reflected by the higher values of the element-centric similarity (ECS) measure. Results are averaged over the 100 subjects, considering the mean between LR and RL phase encoding. For (**a**), a symmetric kernel density estimate has been used for the distributions. Box plots display the median (central line) and interquartile range (IQR). Whiskers extend 1.5 times the IQR, and individual data points, including outliers, are shown with strip plots.

(scaffold scores are omitted for graphical clarity, but they consistently remain below 9%, see SI Fig. S3). This suggests that considering whole-brain connections may include redundant information for brain fingerprinting, while this is mitigated by focusing exclusively on individual functional subnetworks.

To explore subject-specific patterns of brain activation, Fig. 3b reports the coefficient of variation for the triangle approach, when projecting the triangle weights on the cortical brain surface, averaged across the 100 HCP subjects. Interestingly, the interactions between unimodal (i.e., visual, somatosensory) areas and transmodal (e.g., DMN, frontoparietal) brain areas significantly vary between subjects, as indicated by the high values of the coefficient of variation between the seven functional networks. This finding suggests that there might be unique and diverse ways in which individuals' brains engage with areas involved in higher-order cognition (i.e., FP and DMN). For the triangle approach, we also report in SI Fig. S4 the brain activation when considering connections that involve either one (respectively two and three) nodes within one of the seven resting-state functional networks. Comparisons with other methods, including static information-theoretic approaches[40,50] and variants of ICA[67], are provided in the Supplementary Sections S4 and S5, respectively. Although static information-theoretic approaches show a slight improvement over our

temporal topological method when analyzing the entire fMRI scan at once, ICA exhibits lower performance.

## Brain-behavior association

As a third application, we now compare how much the four methods account for variations in cognitive performance. Toward this, we conduct Partial Least Square Correlation (PLSC) analyses between the four static connectivity values (FC, eFC, triangles, scaffold) and ten cognitive scores from the HCP subjects, similar to ref. 68. These cognitive scores encompass ten subdomains within the HCP: episodic memory, executive functions, fluid intelligence, language, processing speed, self-regulation/impulsivity, spatial orientation, sustained visual attention, verbal episodic memory, and working memory[69]. For subdomains with multiple unadjusted raw scores, we obtain a single score by performing principal component analysis on the data, and retaining the first principal component (see "Methods").

It is worth mentioning that PLSC is a multivariate statistical method that identifies latent factors that maximize the covariance between brain patterns (i.e., static connectivity values) and behavioral data, providing insights into which brain regions or networks are most relevant for explaining behavioral variations[70]. See "Methods" for a detailed description of the PLSC approach. In particular, to provide

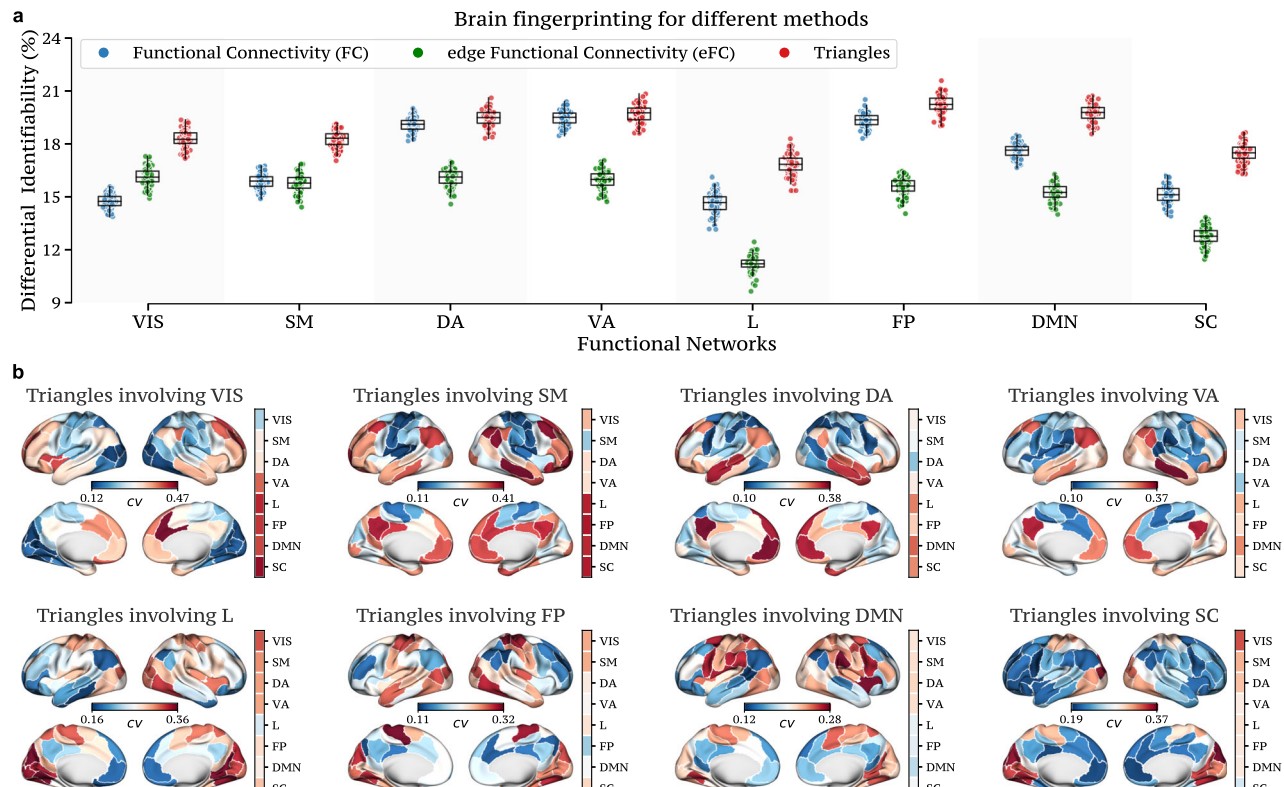

**Fig. 3 | Functional brain fingerprinting performance across methods for fMRI resting-state data. a** For each method, we report the differential identifiability scores obtained when considering the functional connections having at least one node in the functional network analyzed, namely, visual (VIS), somatomotor (SM), dorsal attention (DA), ventral attention (VA), limbic (L), frontoparietal (FP), default mode network (DMN), and subcortical (SC). For graphical clarity, the scaffold method is not reported (<9% for all functional networks). In almost all functional networks, the triangle method outperforms all other approaches. **b** To investigate the patterns of brain activation that are subject-specific, we report the coefficient of variation (cv) for the triangle nodal strength on the cortical brain surface when averaged over the 100 HCP subjects. Interestingly, the way somatosensory areas interact with higher-order networks (i.e., FP and DMN) is quite variable across subjects, as reflected in the high values of the coefficient of variation. Results are obtained by sampling 80 subjects from a total of 100 and repeated this process 100 times (n = 100). The box plots show the median and interquartile range (IQR), with whiskers extending to Q1-1.5IQR and Q3+1.5IQR. Individual data points, including outliers, are shown using scatter plots.

robust scores, we evaluate the covariance explained by significant multivariate correlation components ($p < 0.05$) for the four methods using a bootstrap procedure. We randomly sampled 80 subjects from the total of 100 and repeated this process 100 times.

We started by looking at whole-brain connections. Figure 4a displays the percentage of covariance explained by these significant multivariate correlation components. In this setting, all four methods have similar performances, within the 10–20% range of explained brain-behavior covariance. Notably, when repeating the analyses focusing only on the connections within specific resting-state functional subsystems[8], as illustrated in Fig. 4b, distinctions become noticeable. As we transition from lower-order methods, such as FC or eFC, to higher-order ones, such as triangles or scaffolds, we observe a sharp increase in brain-behavior associations. This is particularly pronounced for somatosensory areas, with triangles reaching ~80% of the covariance explained.

When focusing on the cognitive saliences associated with the significant PLSC components at the local level, we find high variability across methods and functional networks. This indicates that methods beyond functional connectivity tend to explain different cognitive dimensions when focusing on the interactions within functional networks (see Fig. 4c).

To further validate the robustness of this result, Supplementary Section S3 provides additional analyses of the PLSC approach using a limited set of brain features[71]. We also evaluate the performance of methods using the Canonical Correlation Analysis approach[72]. In both cases, at the local level, higher-order methods outperform lower-order approaches for most of the several functional resting-state networks.

## Discussion

Recent conceptual and technical advances in fMRI data analysis provide the opportunity for the field to advance our understanding of brain function and its individual signatures, to accurately predict the emergence of behavior, and to map the complex repertoire of task-induced brain states.

In this study, we focused on the importance of higher-order (group) interactions in fMRI signals, which were inferred using a recently developed topological approach[41]. These initial findings mark an essential milestone in the quest to better comprehend the significance of higher-order interactions in brain dynamics, laying the foundation for further exploration and underscoring the potential of higher-order topological methods in advancing our understanding of human brain function.

We systematically tested this approach on three key and timely brain applications: task decoding, functional brain fingerprinting, and brain-behavior association. To provide a more comprehensive picture of the current state-of-the-art methods, we conducted this assessment in parallel with traditional brain network techniques, such as node[1,2] and eFC[52], with the goal of unraveling the unique advantages offered by higher-order approaches.

Our investigations revealed intriguing patterns. When examining global (whole-brain) higher-order metrics, we found no significant

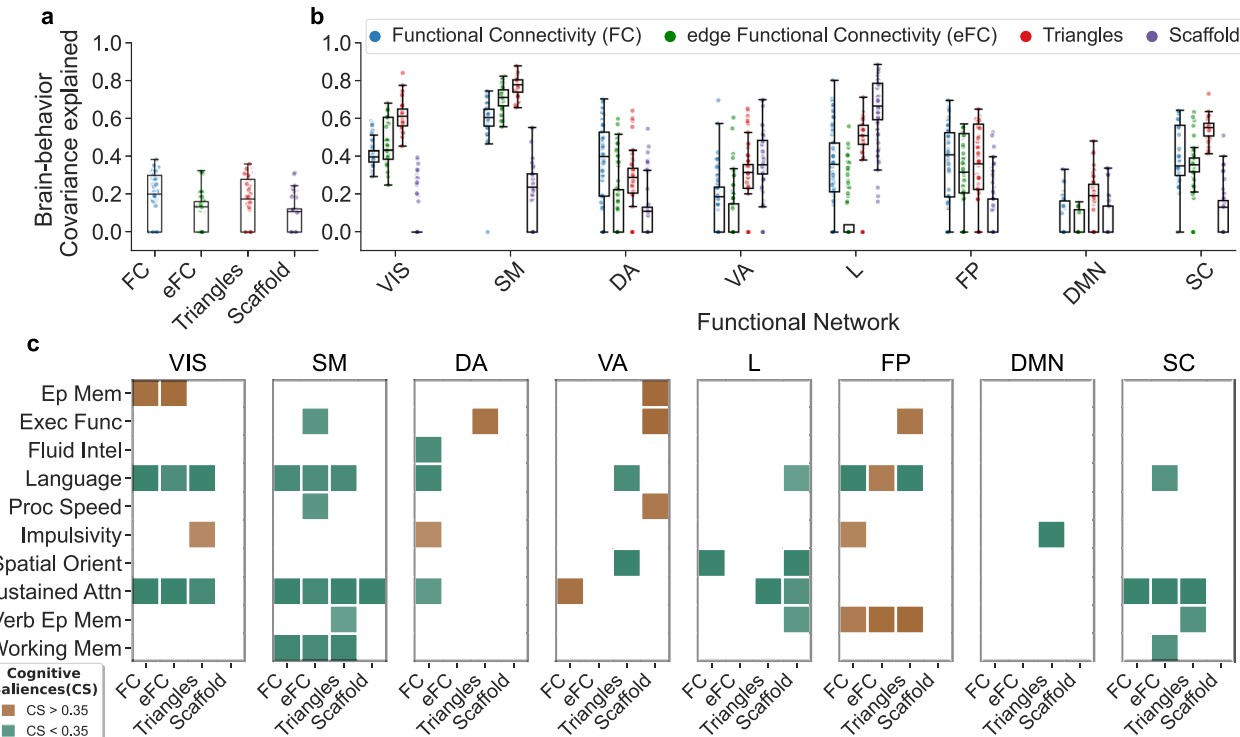

**Fig. 4 | Behavioral significance across methods for fMRI resting-state data.**
**a** Percentage of brain cognition covariance explained by significant multivariate correlation components (permutation testing with *p* < 0.05, 1000 permutations) obtained from Partial Least Square Correlation (PLSC) analyses between the whole-brain functional connections extracted from the different methods and 10 cognitive scores. PLSC components were independently assessed for each method. On a whole-brain level, no differences are noticeable. **b** We repeat the same PLSC analysis considering only the functional connections extracted from the different methods within specific functional networks (i.e., all interactions between nodes of the same network). Remarkably, higher-order methods perform better in terms of covariance explained, reaching 80% in somatosensory areas. **c** For each method and focusing on connections within functional networks, we report the cognitive saliences representing the cognitive domains contributing the most to the brain-cognition multivariate correlation patterns. Colored squares represent cognitive domain weights corresponding to the significant PLSC components. Interestingly, methods beyond functional connectivity tend to explain different cognitive dimensions across the various functional networks. Notice that to obtain a more robust estimate of the covariance explained, we consider a bootstrap procedure randomly sampling 80 subjects from the total of 100 and repeating the PLSC analysis 100 times (*n* = 100). The box plots show the median and interquartile range (IQR), with whiskers extending to Q1-1.5IQR and Q3+1.5IQR. Caps mark the ends of the whiskers. Individual data points, including outliers, are shown using scatter plots. The functional networks analyzed correspond to: visual (VIS), somatomotor (SM), dorsal attention (DA), ventral attention (VA), limbic (L), frontoparietal (FP), default mode network (DMN), and subcortical (SC).

advantages provided by the higher-order approaches. By contrast, when shifting our focus to local indicators or limiting our exploration to resting-state functional subnetworks, higher-order approaches consistently and significantly outperformed classical methods. This suggests that the intricate nuances of brain function are better captured by *local higher-order information*. Specifically, when looking at the temporality of task transitions, we report that local higher-order connectivity features provide a more refined description of the separation between different task-induced brain states. This is in line with previous research on functional "task reconfiguration"[73–80], which showed that the cognitive "switch" between resting-state and task states is more than a general shift in terms of functional links, but rather a complex interplay between maximally distant and minimally distant functional connections. Higher-order approaches provide a novel window of opportunity towards a more refined understanding of centralized and distributed cognitive task processing[81], by quantifying the (instantaneous) level of recruitment of specific functional groups, as well as the dynamical changes in cognitive information processing that specific functional subsystems undergo when the brain wanders across the complex repertoire of human brain states.

Furthermore, local higher-order features provide new insights into individual identification, or brain fingerprinting[54,55]. Our findings showed that local three-way interactions are able to differentiate

individuals more than standard FC-based approaches. This is even more prominent in functional subsystems that are involved in complex cognition (i.e., DMN, FP), immediately followed by somatosensory and visual cortices. Notably, the most identifiable connection involve triangles connecting together these two sensory subsystems with the former ones. This finding suggests that individual identification is a complex interplay between unimodal and transmodal regions, and the proposed higher-order approach allows for a more refined exploration of the interactions between these functional brain communities.

The potential of the approach becomes apparent when looking at the association between brain connectivity features and behavior. One of the ultimate goals of network neuroscience is to extract functional connectome features that could be used for predicting behavior[82]. We provide compelling evidence that higher-order approaches help toward this direction, by reaching brain-behavior covariance peaks of 80% in several functional subsystems, not attainable with standard FC methods, while also covering a broader spectrum of cognitive dimensions. This suggests that large-scale synergistic group interactions might have stronger behavioral relevance than pairwise measures. In fact, while global higher-order indicators fail to yield meaningful insights compared to lower-order methods, local indicators exhibit superior performance in behavioral prediction, surpassing methods relying solely on pairwise interactions. This observation

supports the hypothesis that functional connectivity, when exclusively viewed as bivariate interactions, predominantly captures redundancy-dominated patterns, as highlighted by previous research[24,83]. The presence of higher-order synergies is still vastly unexplored and yet these methodological leaps will be crucial to advance our understanding of the nuanced dynamics of human brain function, and in turn its relationship with the corresponding cognitive and behavioral responses. Future studies should further investigate the relationship between the brain circuits contributing the most to behavioral prediction, to fully elucidate the link between higher-order interaction and brain-behavior associations.

Along these lines, it is noteworthy that while direct measurements of pairwise or group interactions are rarely available[27], topological approaches[84–86] are not the sole means to infer higher-order interactions from brain signals. Recent strides in partial information decomposition[39,40,87] and multivariate information-theoretic approaches have paved the way for analyzing fMRI signals[24,83]. These methods aim to distinguish between redundant and synergistic components of information in systems composed of multiple variables. Notable work by Luppi et al.[23] introduced the concept of a synergistic core within the human brain, where complex processing occurs. Elements of this intricate core, such as the precuneus, prefrontal cortex, and cingulate regions, are partially captured by the scaffold method. These developments highlight the evolving landscape of neuroimaging techniques and emphasize the importance of exploring diverse methodologies to comprehensively unravel the subtleties of brain function. When comparing our approach against multivariate information-theoretic approaches (see SI Section 4)[40,50], we find interesting similarities among activation patterns. That is, violating triangle activation patterns resemble patterns of redundancy, while the homological scaffold aligns more with synergistic contributions. We also remark that, while many existing higher-order methods[24,50] and higher-order connectivity approaches[45–47] require a temporal window for the computation, with few exceptions[22], our topological approach inherently incorporates temporal dynamics, offering diverse scoring options beyond the simple averaging, which was considered in this work.

Our findings outline the substantial improvements that these higher-order methods could bring to the neuroscientific community: here, enabling superior task differentiation, refining functional brain fingerprinting, and establishing more robust associations between brain activity and behavior. This contrast highlights the limitations of relying solely on traditional pairwise or bivariate methods, and the fundamental differences from other computational methods like ICA[48,67]. It advocates for a more nuanced approach, urging researchers to confront the computational challenges posed by higher-order interactions. The true strength of these methods lies in their ability to decipher subtle, localized brain activity patterns that might otherwise remain obscured when using conventional techniques. This emphasizes the significance of embracing advanced methodologies to fully unlock the potential of fMRI data analysis, advancing the understanding of the intricate landscape of brain function.

Overall, we have explored the benefits of the higher-order topological approach in comparison to traditional methods across three key applications: task decoding, human brain fingerprinting, and brain-behavior association. While global indicators struggle to differentiate between various brain dynamics, our research has demonstrated that, despite the increased computational burden, only local higher-order measures offer valuable insights in a range of applications, outperforming other conventional approaches.

A key limitation of our current approach is its computational cost. Analyzing co-fluctuation patterns up to order $k$ results in a time complexity of $\mathcal{O}(N^k)$, which is significantly higher than methods based solely on node (or edge) connectivity. Future research should explore strategies for reconstructing only a subset of the $k$-order interactions.

This could significantly reduce computational complexity, which currently requires around 5 minutes to process data per HCP subject on a powerful workstation (Ryzen Threadripper 3970 with 24 cores).

Furthermore, integrating techniques based on Takens' embeddings, spectral decomposition, or other geometric approaches might offer valuable insights into the causal relationships between brain activity patterns at consecutive time points. These approaches could potentially complement our current method and provide a more comprehensive understanding of brain network dynamics.

## Methods

### HCP dataset and functional preprocessing

The fMRI dataset used in this work is from the Human Connectome Project (HCP, www.humanconnectome.org), Release Q3. We considered the 100 unrelated subjects (54 females and 46 males, mean age = 29.1 ± 3.7 years) as provided from the HCP 900 subjects data release[64,65]. This subset of subjects provided by HCP ensures that they are not family relatives. All experiments were reviewed and approved by the local institutional ethical committee (Swiss Ethics Committee on research involving humans). Informed consent forms, including consent to share de-identified data, were collected for all subjects (within the HCP) and approved by the Washington University Institutional Review Board. All methods were carried out in accordance with relevant guidelines and regulations. This criterion was crucial to exclude family-structure co-variables in our analyses, as well as possible behavioral confounds. Per HCP protocol, all subjects gave written informed consent to the HCP consortium. The fMRI resting-state runs (HCP filenames: rfMRI_REST1 and rfMRI_REST2) were acquired in separate sessions on two different days, with two different acquisitions (left to right or LR and right to left or RL) per day[10,59]. For all sessions, data from only the LR phase-encoding run was used to calculate the HOI structures and average across different days (one LR REST1, one LR REST2) per subject. The seven fMRI tasks were the following: gambling (tfMRI_GAMBLING), relational (tfMRI_RELATIONAL), social (tfMRI_SOCIAL), working memory (tfMRI_WM), motor (tfMRI_MOTOR), language (tfMRI_LANGUAGE, including both a story-listening and arithmetic task) and emotion (tfMRI_EMOTION). The working memory, gambling and motor task were acquired on the first day, and the other tasks were acquired on the second day[65,69]. For this study, we used the minimally preprocessed HCP resting-state data[59], with the following preprocessing steps. First, we applied a standard general linear model regression that included detrending and removal of quadratic trends; removal of motion regressors and their first derivatives; removal of white matter, cerebrospinal fluid signals, and their first derivatives; and global signal regression (and its derivative). Second, we bandpass-filtered the time series in the range of 0.01 to 0.15 Hz. Last, the voxel-wise fMRI time series were averaged into their corresponding brain nodes of the Schaefer 100 atlas[58] (as described in the next section) and then z-scored.

### Brain atlas

We used a cortical parcellation of 100 brain regions as recently proposed by Schaefer and collaborators[58]. For completeness, 16 subcortical regions and 3 cerebellar regions were also added, as provided by the HCP release (filename "Atlas_ROI2.nii.gz"), resulting in a final brain atlas of 119 brain nodes.

### Classical methods and topological higher-order organization of fMRI signals

The classical approach in network neuroscience to construct networks from fMRI data requires estimating the statistical dependency between every pair of time series[2,88,89]. The magnitude of that dependency is usually interpreted as a measure of how strongly (or weakly) those parcels (or voxels) are functionally connected to each other. By far the

most common measure of statistical dependence is the Pearson correlation coefficient. Let us consider a $N$-dimensional real-valued time series $\{\mathbf{x}(t)\}_{i=1}^{N}$ with $T$ time points, where $\mathbf{x}_i = [x_i(1), x_i(2), ..., x_i(T)]$ represents the generic time series recorded from parcel (or voxel) $i$. In the context of fMRI, this is typically referred to as *(BOLD) time signal*. The classical approach to compute Pearson's functional connectivity (FC) matrix relies on computing the correlations between all the possible pairs of parcels $\mathbf{x}_i$ and $\mathbf{x}_j$. If there are $N$ nodes, the FC matrix has dimensions $N \times N$. Following the edge-centric approach proposed in ref. 52, however, it is possible to estimate the instantaneous co-fluctuation magnitude between a pair of time series $\mathbf{x}_i$ and $\mathbf{x}_j$ -once they have been $z$-scored- by estimating their element-wise product. That is, for every pair of time series, a new time series encodes the magnitude of co-fluctuation between those signals resolved at every moment in time, the so-called *edge time signal*. In a similar fashion, as done for FC, it is possible to compute the correlations between all pairs of edge time signals. With $N$ parcels, this results in a matrix, the *eFC*, with dimension $\binom{N}{2} \times \binom{N}{2}$.

For the topological higher-order approach[41], we generalize the concept of edge-time series to the case of higher-order interactions, i.e., triangles, tetrahedron, etc. We first $z$-score each original time series $\mathbf{x}_i$, such that $\mathbf{z}_i = \frac{x_i - \mu[\mathbf{x}_i]}{\sigma[\mathbf{x}_i]}$, where $\mu[\bullet]$ and $\sigma[\bullet]$ are the time-averaged mean and standard deviation. We then calculate the generic element at time $t$ of the $z$-scored $k$-order co-fluctuations between $(k + 1)$ time series as

$$\xi_{0...k}(t) = \frac{\prod_{p=0}^{k} z_p(t) - \mu\left[\prod_{p=0}^{k} \mathbf{z}_p\right]}{\sigma\left[\prod_{p=0}^{k} \mathbf{z}_p\right]}, \qquad (1)$$

where also in this case $\mu[\bullet]$ and $\sigma[\bullet]$ are the time-averaged mean and standard deviation functions. In order to differentiate concordant group interactions from discordant ones in a $k$-order product, concordant signs are always positively mapped, while discordant signs are negatively mapped. That is,

$$\text{sign}\left[\xi_{0...k}(t)\right] := \qquad (2)$$

where sgn[•] is the signum function of a real number. Formally, the weight $w_{0...k}(t)$ of the $k$-order co-fluctuations at time $t$ is defined as:

$$w_{0...k}(t) = \text{sign}[\xi_{0...k}(t)]|\xi_{0...k}(t)| \qquad (3)$$

In this way, all the perfectly coherent contributions (namely, either all positive or all negative) will be mapped as positive. This is because in our method we are mainly focusing on the coherent contributions within the multivariate time series. If we compute all the possible products up to order $k$, this will result in $\binom{N}{k+1}$ different co-fluctuation time series for each order $k$.

For each time $t$, we condense all the different $k$-order co-fluctuations into a weighted simplicial complex $\mathcal{K}^t$. Formally, a $(d-1)$-dimensional simplex $\sigma$ is defined as the set of $d$ vertices, i.e., $\sigma = [p_0, p_1, ..., p_{d-1}]$. A collection of simplices is a simplicial complex $\mathcal{K}$ if for each simplex $\sigma$ all its possible subfaces (defined as subsets of $\sigma$) are themselves contained in $\mathcal{K}$[90]. Weighted simplicial complexes are simplicial complexes with assigned values (called weights) on the simplices. In a nutshell, this provide a time-resolved summary of the coherent (and decoherent) landscape of group interactions present in the fMRI data employed.

As done in ref. 41, also in this work we only consider co-fluctuations of dimension up to $k = 2$, so that triangles represent the only higher-order structures in the weighted simplicial complex $\mathcal{K}$, and weights on the simplices, i.e., $w_{ij}$ and $w_{ijk}$, represent the

magnitude of edges and triangles co-fluctuations. We note that while the concept of edge time series represents a temporal unwrapping of the Pearson's correlation coefficient[51], the generalization to $k$-order co-fluctuations does not correspond to any specific statistical measure. However, the sign remapping emphasizes the purely coherent co-fluctuations, allowing us to better focus and discriminate these contributions.

## Hyper coherence and hyper complexity

To examine the structure of the weighted simplicial complex $\mathcal{K}^t$ across different scales, we rely on persistent homology, which is a recent technique in computational topology that has been largely used for the analysis of high dimensional datasets[56,91] and in disparate applications[17,92,93]. The key idea involves constructing a sequence of successive simplicial complexes that progressively approximate the original weighted simplicial complex with greater precision. This sequence of simplicial complexes, i.e., $\emptyset = \mathcal{S}_0 \subset \mathcal{S}_1 \subset ... \subset \mathcal{S}_l \subset ... \subset \mathcal{S}_n$, is such that $\mathcal{S}_i \subset \mathcal{S}_j$ whenever $i < j$ and is called a filtration. In our context, we build a filtration through the following steps:

- Sort the weights of the links and triangles in descending order: the parameter $\epsilon_l \in \mathbb{R}$ serves as the scanning parameter in the sequence. Equivalently, $\epsilon_l$ is the parameter that keeps track of the actual weight as we gradually scroll the list of weights.
- At each step $l$, remove all triangles that do not satisfy the simplicial closure condition, i.e., $\exists \, !(i, j): w_{ij} < w_{ijk}$. These violating triangles, considered as violations, are added, along with their corresponding weights, in the *list of violations* $\Delta_v = \{(i, j, k), w_{ijk}\}$. The remaining links and triangles with a weight larger than $\epsilon_l$ constitute the simplicial complex $\mathcal{S}_l$.

We define the *hyper coherence indicator*, as the fraction of violating coherent triangles (i.e., violating triangles with a weight greater than zero) over all the possible coherent triangles (i.e., triangles with a weight greater than zero).

Persistent homology examines topological structure changes throughout the filtration $\mathcal{S}_l$, offering a natural measure of robustness for emerging topological features across various scales. Specifically, it is possible to keep track of these topological changes by looking at each $k$-dimensional cycle in the homology group $H_k$. In our case, we focus on the 1-dimensional holes (i.e., loops) within the homology group $H_1$. At each filtration step, a generator $g$ uniquely identifies a 1-dimensional cycle by its constituting elements. The significance of the 1-dimensional hole $g$ is encoded through "time-stamps" representing its birth $b_g$ and death $d_g$ along the filtration $\{\mathcal{S}_l\}$[18]. The persistence $\pi_g = d_g - b_g$ defines the lifespan of the one-dimensional cycle $g$, providing insights into its importance.

The results of persistent homology group $H_1$ are commonly visualized through points in the two-dimensional persistence diagram. In this diagram, each point $(b_g, d_g)$ represents a one-dimensional hole $g$ appearing across the filtration. As a result, this diagram provides a compressed summary describing how long 1D cycles live along the filtration and can be used as a proxy of the "complexity" of the underlying space. In particular, we define the *hyper complexity indicator* as the (sliced) Wasserstein distance[94] between the persistence diagram of $H_1$ and the empty persistence diagram, corresponding to a space with trivial $H_1$ homology. This provides us with a measure of the topological complexity of the landscape of coherent and incoherent co-fluctuations.

## Homological scaffold

For a finer description of the topological features present in the persistence diagram, we rely on the frequency homological scaffold proposed in ref. 18. This object is a weighted network comprising of all cycle paths corresponding to generators $g_i$, weighted by the number of cycles to which it belongs. In other words, if an edge $e$ belongs to

multiple 1-dimensional cycles $g_0, g_1, ..., g_s$, its weight $\bar{w}_e^f$ is weighted by the number of different cycles it belongs to, i.e.,:

$$\bar{w}_e^f = \sum_{g_i} 1_{e \in g_i}$$

where $1_{e \in g_i}$ is the indicator function for the set of edges composing $g_i$. The information provided by the homological scaffold allows us to decipher the role that different links provide regarding the homological properties of the system. A large weight $\bar{w}_e^f$ for a link $e$ implies that such a link acts as a locally strong bridge in the space of coherent and decoherent co-fluctuations[18].

### Functional brain fingerprinting

Motivated by recent research focused on maximizing connectivity fingerprints in human functional connectomes[54,55,66], we investigated fMRI inter-subject identifiability across four different methods using static approaches. To quantify this, we rely on the "identifiability matrix" $\mathbf{A}$[55], a square and non-symmetric similarity matrix of size $S$, with $S = 100$ representing the number of subjects in the dataset. This matrix captures information regarding each subject's self-similarity during test and retest sessions ($I_{self} = \langle a_{ii} \rangle$ for main diagonal elements) and the similarity between subjects ($I_{others} = \langle a_{ij} \rangle$ for off-diagonal elements). The similarity between two connectivity measures (i.e., functional connectivity, eFC, average violating triangles, average scaffold) is quantified using Pearson's correlation coefficient based on the test/retest connectivity measures. The discrepancy between $I_{self}$ and $I_{others}$ (referred to as "Differential Identifiability" - $I_{diff}$) serves as a score reflecting the fingerprinting potential of a given dataset[55].

### Partial Least Square Correlation (PLSC)

To examine the relationship between brain activity and behavior across the four different methods, we rely on Partial Least Square Correlation (PLSC) analyses. These analyses aimed to find connections between the functional links of each method and 10 cognitive scores across subjects. For the whole-brain connections, we consider a different number of functional connections for each method: $\binom{N}{2}$ for functional connectivity (the upper triangular part of the functional connectome), $\binom{\binom{N}{2}}{2}$ for eFC (the upper triangular part of the eFC), $\binom{N}{3}$ for average violating triangles, and $\binom{N}{2}$ for scaffold. To enhance the stability of the functional connections, we averaged them over the two resting-state fMRI recordings (i.e., $REST_1$ and $REST_2$). The cognitive scores considered belong to 10 cognitive subdomains in the HCP dataset, covering behavioral traits like episodic memory, executive functions, fluid intelligence, language, processing speed, self-regulation/impulsivity, spatial orientation, sustained visual attention, verbal episodic memory, and working memory[69]. When multiple raw scores were available for a subdomain, we obtained a single score through principal component analysis.

We performed the PLSC analysis separately for each of the four methods. PLSC identifies linear combinations of functional connectivity values that maximally covary with linear combinations of cognitive scores, determined through singular value decomposition of the data covariance matrix[70]. The weights of these linear combinations are commonly referred to as brain function and cognitive saliences, representing the left and right singular vectors of the data covariance matrix. We assessed the statistical significance of the PLSC components using permutation testing (1000 permutations; correlation patterns with $p < 0.05$ were considered significant)[70]. To evaluate the reliability of non-zero salience values, we implemented a

bootstrapping procedure (1000 random data resampling with replacement) and calculated standard scores with respect to the bootstrap distributions (salience values were considered reliable if the absolute standard score >2)[70,95] We quantified the amount of cognitive traits' variance explained by functional connectivity values by summing the squared singular values corresponding to the significant PLSC components. This sum was normalized by the total sum of squared singular values obtained for each PLSC analysis[70]. For a more robust estimation of the covariance explained by significant multivariate correlation components ($p < 0.05$) for the four methods, we employed a bootstrap procedure. This involved randomly subsampling 80 subjects from a total of 100 and repeating the PLSC analysis 100 times. The reliability of non-zero salience values is obtained considering those scores that are reliable (i.e., absolute standard score > 2) for more than 60% of the total number of bootstrap sub-samples.

### Reporting summary

Further information on research design is available in Nature Portfolio Reporting Summary linked to this article.

## Data availability

The dataset analyzed during the current study, i.e., the Human Connectome Project, Release Q3, is available in the Human Connectome Project repository (http://www.humanconnectome.org/). The pre-processed resting-state fMRI data, as well as the concatenated task fMRI data, are available on Zenodo https://zenodo.org/records/13786223.

## Code availability

The code to reproduce the main results of this study is available at https://zenodo.org/records/13786134. For a maintained version of the code, please refer to AS's GitHub repository https://github.com/andresantoro/Brain_HORS.

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

## Acknowledgements
Data were provided by the Human Connectome Project, WU-Minn Consortium (principal investigators: David Van Essen and Kamil Ugurbil; 1U54MH091657) funded by the 16 NIH Institutes and Centers that support the NIH Blueprint for Neuroscience Research; and by the McDonnell Center for Systems Neuroscience at Washington University. The authors thank Dr. J. Goñi, Dr. D. Van De Ville, Dr. N. Gninenko and M. Capatonda for insightful discussions. E.A. acknowledges financial support from the SNSF Ambizione project "Fingerprinting the brain: Network science to extract features of cognition, behavior and dysfunction" (grant number PZ00P2_185716). A.S. and E.A. acknowledge support from SNSF COST project "Mapping the higher-order dynamics of neurodegeneration in human brain networks" (grant number IZCOZO_198144). The funders had no role in study design, data collection and analysis, decision to publish, or preparation of the manuscript.

## Author contributions
A.S., F.B., M.L., G.P. and E.A. conceptualized the study. A.S. developed the code, performed the numerical analysis, prepared the figures, and wrote the original draft of the manuscript. All authors contributed to the review and editing of the draft.

## Competing interests
The authors declare no competing interests.
