## [Transparent Peer Review file · Nature Communications]

Higher-order connectomics of human brain function reveals local topological signatures of task decoding, individual identification, and behavior

Corresponding Author: Professor Enrico Amico

Version 0:

Reviewer comments:

Reviewer #1

(Remarks to the Author)

The paper presents an advancement in the study of human brain activity by focusing on higher-order interactions (HOIs) within fMRI data. Traditional models, which emphasize pairwise interactions between brain regions, are expanded upon by investigating the impact and utility of HOIs that involve three or more regions. Through a comprehensive analysis of HCP fMRI time series, the authors demonstrate that higher-order approaches not only enhance the decoding of dynamic changes across various tasks but also improve the identification of individuals based on unimodal and transmodal functional subsystems. Furthermore, these approaches strengthen the associations between brain activity and behavior, offering new insights into the complex organization of the human brain beyond what is possible with traditional pairwise models.

The paper is very topical with an increasing interest beyond pairwise methods. It is also very well written.

Major comments

- Can you please explain what novel contribution of your work within the context of the state of the art (see papers below, some of the other mentioned in Discussion). What topological data analysis methods employed here (edge, triangle, scaffolds etc) contribute over and above the previous work?

Comparing the performance of your TDA based analysis with one or two most notable methods would enhance the impact of the paper.

<https://www.sciencedirect.com/science/article/pii/S0969996122003102>

<https://www.ncbi.nlm.nih.gov/pmc/articles/PMC10125999/>

<https://www.sciencedirect.com/science/article/pii/S0896627321007157>

- The paper touches upon the potential of other methodologies, such as partial information decomposition and multivariate information-theoretic approaches, to infer higher-order interactions. Can you provide some insights into how these other methods compare/contrast with the TDA based methods you propose?

- The discussion acknowledges the increased computational burden of higher-order approaches. While these methods offer valuable insights, their practical application will be perhaps limited by the computational challenges they pose. Addressing these challenges will be crucial for making higher-order analyses more accessible and feasible in broader research contexts. Could you please provide some quantitative indication of the complexity of calculations required compared to the simpler pair-wise methods?

- Authors used the PLSC approach for relating brain and behaviour. The more common / generalized multivariate approach is canonical variate analysis. Please elaborate how LSC differ and why it is preferred here and compare the two methods performance.

Reviewer #2

(Remarks to the Author)

Summary

In this study, Santoro et al., use a time-resolved, higher-order analysis (recently introduced by the same authors) to explore whether higher-order interactions provide a significant improvement over classic, pairwise functional connectivity analyses. To do this, they run a battery of tests, including inter-scan subject fingerprinting, task-decoding, and brain-behavior association correlations. They show evidence that the higher-order measures generally out-perform the lower-order ones (although with caveats). This is a significant finding for the nascent field of higher-order network neuroscience – many studies have been published using fancy methods to show that higher-order interactions exist, but it's generally been unclear whether it's all worth the effort. Santoro et al., show strong evidence that it is worth the effort. I think that the paper is strong and generally well written – I see no glaring issues and I am happy to accept it after minor revisions (generally things that I wish were explained better).

Comments (in no particular order)

- I am struggling to understand how to interpret these higher-order edge time series. I get that for pairs of regions, the ETS is essentially a local decomposition of the Pearson product-moment coefficient (Betzel et al., showed this, if memory serves). However, there is no general N-way generalization of the Pearson coefficient, so I'm not entirely sure what the product of $x_1 * x_2 * \dots * x_N$ really means. I would like the authors to spend some time meditating on this – if the construction is just a mathematical “trick” to juice the performance of the methods then that's fine (statistics is full of such things), but it is a little bit unsatisfying as-is.

- Naively, shouldn't the parity be based on sum of signs modulo 2? For example, $-1 \times -1 \times 1$ has parity of 1, but in your schema it would be assigned a negative sign? Am I understanding that right? The only moments that get a positive score are either all negative or all positive? The product of the z-scores is already a signed value – why change it?

- Have you considered other ways of tracking the interactions of the elements? For example, for each TR, the instantaneous z-scores define a vector: perhaps you could use the Manhattan norm, Euclidean norm, Chebyshev norm, etc? Since a k-dimensional time series is a trajectory through k-d space, taking the norm at each point seems more interpretable to me.

- In the same vein: if you have k time series and use element-wise multiplication to compress them down to a 1-dimensional time series, haven't you *lost* a lot of information about the interaction? The move from a k-dimensional space to a 1-dimensional space is necessarily degenerate: there are many sets of k numbers that will produce the same product, so you've lost a lot of information in the joint-state of the k time series (which, to my mind, seems like it would be important for a higher-order analysis). Is this an analysis of higher-order redundancy or higher order synergy?

- Why do ETS perform so much worse than either FC or HOI? Since the ETS is basically an “unrolling” of the pairwise correlation, it seems like everything available in the FC should also be available in the ETS?

- I'm not sure what this sentence means:

"While recent technologies now allow us to record higher-order phenomena at the micro-scale neuronal level (e.g., the simultaneous firing of groups of neurons) [35–37], such data remains unavailable to explore large-scale brain dynamics."

What does "unavailable" mean? Not shared? Or we don't have the tools to get at the HOIs?

- Small stylistic thing: the transition into "While recent edge-centric..." is a bit jarring.

-

Reviewer #3

(Remarks to the Author)

This is an interesting and generally well-written paper. The report brings together number of technical advancements in fMRI. Ultimately however, it often feels the “whole is less than the sum of its parts”. While it is exciting to see recent extensions of edge time series further extended, it remains a bit unclear how these methods compare to other for example high dimensional functional data analysis techniques (e.g., ICA) for the reviewed use cases or what the main take aways are from these different analyses that often lead to mixed evidence. While this alone isn't a problem writ large with the analyses here, a lack of clarity on alternative high dimensional functional data techniques here for certain analyses (e.g., prediction of phenotypes, fingerprinting/differential heritability) makes it difficult to determine whether just more complex representations of data are outperforming less complex ones, or whether this particular framework has additional purchase. Additional specific comments are listed below:

Theme 1. Framing and level of complexity.

A. Its is difficult to determine for the reader in the intro the throughline the connects the conducted analyses. As stated above, an extension of edge time series seems interesting but most of the included analyses focus on use cases outside of network neuroscience or functional neuroanatomy, but rather prediction (behavior, task decoding). This is okay in theory, but the

general premise can be a bit hard to follow. Are the authors suggesting this should be how new investigators analyzed fMRI data in prediction contexts? Much more context would be helpful if that were the case and if not to better clarify what exact value add these analyses have?

B. The introduction is written generally very well for a broad audience. That said, connecting the dots here and keeping the high-level concepts clear would be important for a broad audience at this outlet.

Theme 2: Lack of comparisons to other functional data analysis techniques given the lack of clarity on the goal (see above). This likewise extends to the methods used for prediction of phenotypes and decoding.

Theme 3: Magnitude/scaling of brain-behavior phenomena.

A. It is a bit unclear as to what exact scaling is happening in Figure 4 but these values are very very high. Higher than predictions from multiple independent groups for these same types of behaviors (e.g. Marek et al., Nature, 2022; Chen et al., Nature Communications, 2022; Schulz et al., Cell Reports, 2024) each of which consider multiple methods. Can the authors clarify why this is?

B. Also the models perform worse when including more brain regions which likely suggests poorly condition whole brain models.

Version 1:

Reviewer comments:

Reviewer #1

(Remarks to the Author)

I am very pleased to see the revised manuscript in which authors have thoroughly addressed all the concerns I raised. The paper is now very much improved and I would read it again once it is in print.

Adeel Razi
Monash University
Australia

Reviewer #2

(Remarks to the Author)

The authors have addressed my points. I am satisfied with the paper as it currently stands.

Reviewer #3

(Remarks to the Author)

I commend the authors on a thoughtful revision. I would encourage a final pass (though leave this to the editor) to ensure the scaling and transparency of out-of-sample brain-behavior magnitude predictions is accurate and faithful to the methods used. I note this in the context of large scale debates regarding brain-behavior reproducibility, overfitting, and cross-validation scenarios.

Topological brain connectomics for task classification, brain fingerprinting,
and behavioral association
(NCOMMS-23-62931-T)
Reply to the Reviewers

June 8, 2024

General reply to the reviewers

Dear reviewers,

Thank you for the thorough and insightful review of our manuscript. We are grateful for the time and attention paid to our work. We sincerely want to thank you for your attentive comments and valuable advice, which have been all taken into consideration. We feel that our manuscript came out stronger from this constructive review process.

We provide below a list of the modifications made, followed by in-depth replies to all comments. We also provide an annotated copy of the main text, where the modifications are highlighted.

We thank you again for your time and insights.

Yours sincerely,

The authors

Detailed reply to the report of reviewer # 1

COMMENT 1.0

The paper presents a advancement in the study of human brain activity by focusing on higher-order interactions (HOIs) within fMRI data. Traditional models, which emphasize pairwise interactions between brain regions, are expanded upon by investigating the impact and utility of HOIs that involve three or more regions. Through a comprehensive analysis of HCP fMRI time series, the authors demonstrate that higher-order approaches not only enhance the decoding of dynamic changes across various tasks but also improve the identification of individuals based on unimodal and transmodal functional subsystems. Furthermore, these approaches strengthen the associations between brain activity and behavior, offering new insights into the complex organization of the human brain beyond what is possible with traditional pairwise models.

The paper is very topical with an increasing interest beyond pairwise methods. It is also very well written.

Reply 1.0: Thank you for judging the topic of our manuscript timely and forward-looking. We are also very grateful for the positive general overview of our work.

COMMENT 1.1

Can you please explain what novel contribution of your work within the context of the state of the art (see papers below, some of the other mentioned in Discussion). What topological data analysis methods employed here (edge, triangle, scaffolds etc) contribute over and above the previous work?

Comparing the performance of your TDA based analysis with one or two most notable methods would enhance the impact of the paper.

<https://www.sciencedirect.com/science/article/pii/S0969996122003102>

<https://www.ncbi.nlm.nih.gov/pmc/articles/PMC10125999/>

<https://www.sciencedirect.com/science/article/pii/S0896627321007157>

Reply 1.1: We thank the referee for this comment. In the revised version of the manuscript, we have expanded our discussion to include a more detailed paragraph on the benefits of employing topological data analysis when compared to other state-of-the-art techniques, as the ones mentioned by the referee. In particular, we emphasize the importance of the temporality in our approach, alongside other relevant factors, such as the presence of local and global indicators, which are not found in other approaches.

Moreover, in the revised version of the Supplementary Material, we have included two new plots comparing the performance of higher-order information theoretic measures in brain fingerprinting and brain-behavior association. Our findings illustrate varying performances for the information-theoretic approaches, excelling in brain fingerprinting but lagging in brain-behavior association. However, it is important to note that these information-theoretic methods require long temporal windows for their computation, whereas our proposed approach inherently incorporates temporal dynamics, offering diverse scoring options beyond the simple averaging (the one considered in the manuscript).

Action taken 1.1: We reshaped the discussion in the main text to emphasize the advantages of our temporal topological data analysis (TDA) approach compared to other state-of-the-art methods. Additionally, Supplementary Section S4 now presents a comparison between higher-order information theoretic measures (O- and S- information, Dual Total Correlation) and the other approaches originally discussed in the manuscript. The results underscore the distinct information yielded by each method and for different brain applications.

For completeness, we report below the new paragraphs in the main text and in Supplementary Sections. New text is highlighted in red.

DISCUSSION:

When comparing our approach against multivariate information-theoretic approaches (see SI Section 4) [40,50], we find interesting similarities among activation patterns. That is, violating triangle activation patterns resemble patterns of redundancy, while the homological scaffold aligns more with synergistic contributions. We also remark that, while many existing higher-order methods [24,50] and higher-order connectivity approaches [45-47] require a temporal window for the computation, with few exceptions [22], our topological approach inherently incorporates temporal dynamics, offering diverse scoring options beyond the simple averaging, which was considered in this work.

SUPPLEMENTARY SECTION S4 — COMPARISON WITH MULTIVARIATE INFORMATION-THEORETIC APPROACHES

In this section, we compare the topological approach presented in the main text with recent static higher-order information-theoretic approaches based on O-information, S-information, and Dual Total Correlation (DTC) [8]. We relied on the code provided in Ref. [6] for this comparison.

We evaluated the similarity between the activation patterns of local higher-order measures (violating triangles and the homological scaffold) identified by our topological approach and those identified by the information-theoretic approaches. We remark a crucial difference between the methods: Our topological approach inherently captures temporal dynamics, therefore we need to average the activation patterns across time for each subject. By contrast, the information-theoretic approaches are computed over the entire window of the fMRI scan (1200 fMRI volumes).

S4.1 Cortical profiles

Figure S8 shows the results of our comparison. We projected the violating triangles, homological scaffold, O-information, and S-information onto the cortical surface (Schaefer 100), as shown in panels **a-f**. We found a moderate correlation (Pearson's correlation coefficient ≈ 0.42) between the topological activations and those from the information-theoretic methods. The correlation values vary depending on whether we consider activation at the nodal level (after projection) or at the original triangle level (panels **g** and **h**).

Furthermore, brain surface projections of the scaffold-to-triangles ratio and the redundancy-to-synergy gradient scores (based on their respective ranks) also revealed similar cortical patterns ($\rho = 0.41$). Notice that the redundancy and synergy contributions are derived from the O-information vector of size $\binom{N}{3}$ isolating only the positive and negative contributions, respectively. In simpler terms, in resting-state fMRI data, we find that the violating triangle activation pattern resembles redundancy, while the homological scaffold aligns more with synergistic contributions.

S4.2 Functional brain fingerprinting

This section compares the performance of information-theoretic methods against our topological approach in the context of functional brain fingerprinting. Similar to the analysis in Figure 3 of the main text, we assessed the differential identifiability achieved by each method. Figure S9 shows the results when considering connections involving at least one node within one of the seven resting-state functional networks. Interestingly, the information-theoretic methods outperform the topological methods for all networks except the visual network. This is remarkable, but it could be due to the fact that information-theoretic approaches are inherently static, i.e. they analyze the entire 1200 fMRI window at once. In contrast, our method obtains activation vectors by simply averaging across the activation of the individual 1200 volumes, potentially introducing noise or artefacts.

S4.3 Brain-behavior association

This section compares the performance of information-theoretic methods with our topological approach in uncovering brain-behavior associations, using the PLSC technique. Similar to the analysis in Figure 4 of the main text, we assessed the amount of covariance explained by each method, both globally and locally. Figure S10a shows the percentage of total covariance explained by the significant multivariate correlation components when considering all brain connections. In this whole-brain analysis, O-information yielded the highest brain-behavior covariance, with an average value across bootstrapped runs of around 40%. However, when we analyze connections within specific resting-state functional networks (Fig. S10b), the results differ. With a few exceptions, such as the dorsal-attention (DA) and frontoparietal (FP) networks, triangles and scaffold measures consistently explained a higher percentage of covariance for brain-behavior associations.

Figure S8: Cortical profiles of topological and information-theoretic measures. We report the activation of the local higher-order measures, namely (a) violating triangles and (b) homological scaffold, and those identified by the information-theoretic approaches, (d) O-info and (e) S-info, respectively. Brain surface projections of the (c) scaffold-to-triangles ratio and the (f) redundancy-to-synergy gradient scores (based on their respective ranks) also reveal similar patterns, as confirmed by the Pearson’s correlation coefficient of $\rho = 0.42$. We also compare between the topological activations and the corresponding maps obtained from the information-theoretic methods, either at the (g) nodal level (after projection) or at the (h) original triangle level.

Figure S9: Functional brain fingerprinting performance across methods for resting-state fMRI data. For each method, we present the differential identifiability scores for functional connections associated with the following networks: visual (VIS), somatomotor (SM), dorsal attention (DA), ventral attention (VA), limbic (L), frontoparietal (FP), and default mode network (DMN). Except for the visual network, the S-info/DTC metric outperforms all other approaches.

Figure S10: Behavioral significance across methods for fMRI resting-state data. (a) Percentage of brain cognition covariance explained by significant multivariate correlation components ($p < 0.05$) obtained from PLSC analyses between the whole-brain functional connections extracted from the different methods and 10 cognitive scores. PLSC components were independently assessed for each method. To obtain a more robust estimate of the covariance explained, we consider a bootstrap procedure randomly subsampling 80 subjects from the total of 100 and repeating the PLSC analysis 100 times. On a whole-brain level, O-information performs better than all the other approaches. (b) We repeat the same PLSC analysis considering only the functional connections extracted from the different methods within specific functional networks (i.e. all interactions between nodes of the same network). As shown in Figure 4 of the main text, also in this case higher-order methods perform better in terms of covariance explained, reaching 80% in somatosensory areas. Functional networks: visual (VIS), somatomotor (SM), dorsal attention (DA), ventral attention (VA), limbic (L), frontoparietal (FP), and default mode network (DMN). Except for the visual network, the S-info/DTC metric outperforms all other approaches.

COMMENT 1.2

The paper touches upon the potential of other methodologies, such as partial information decomposition and multivariate information-theoretic approaches, to infer higher-order interactions. Can you provide some insights into how these other methods compare/contrast with the TDA-based methods you propose?

Reply 1.2: A key difference of the TDA approach lies in its inherent ability to capture the temporal dynamics. This stands in contrast to most other existing methods (with a few exceptions, e.g. L. Faes et al (2022). IEEE Transactions on Signal Processing, 70, 5766-5777), which typically rely on a temporal window for calculations. Consequently, the TDA approach allows for the computation of *temporal* global and local indicators.

In addition to the comparison reported in Comment 1.1, to further explore the differences between redundancy/synergy approaches and TDA, we compared them on static resting-state fMRI data. Specifically, we projected violating triangles and the homological scaffold onto the cortical surface. We found a moderate correlation (Pearson's correlation $\rho \approx 0.42$) with the corresponding maps obtained from redundancy and synergy analyses, respectively. In other words, the violating triangle activation pattern moderately resembles redundancy, while the homological scaffold aligns more with synergistic contributions.

Action taken 1.2: In the revised version of the manuscript, we have included a new paragraph in the discussion emphasizing the temporal significance of our method compared to other approaches that infer higher-order interactions. Additionally, we have conducted a new analysis by assessing the correlation between the TDA local indicators and redundancy/synergy indicators when evaluated at the cortical level using resting-state fMRI data. We reported it in Supplementary Section S4.1.

For reference, the new paragraphs in the main text are provided below (but the same as in the previous comment). New additions are highlighted in red.

DISCUSSION:

When comparing our approach against multivariate information-theoretic approaches (see SI Section 4) [40,50], we find interesting similarities among activation patterns. That is, violating triangle activation patterns resemble patterns of redundancy, while the homological scaffold aligns more with synergistic contributions. We also remark that, while many existing higher-order methods [24,50] and higher-order connectivity approaches [45-47] require a temporal window for the computation, with few exceptions [22], our topological approach inherently incorporates temporal dynamics, offering diverse scoring options beyond the simple averaging, which was considered in this work.

SUPPLEMENTARY SECTION S4 — COMPARISON WITH MULTIVARIATE INFORMATION-THEORETIC APPROACHES

In this section, we compare the topological approach presented in the main text with recent static higher-order information-theoretic approaches based on O-information, S-information, and Dual Total Correlation (DTC) [8]. We relied on the code provided in Ref. [6] for this comparison.

We evaluated the similarity between the activation patterns of local higher-order measures (violating triangles and the homological scaffold) identified by our topological approach and those identified by the information-theoretic approaches. We remark a crucial difference between the methods: Our topological approach inherently captures temporal dynamics, therefore we need to average the activation patterns across time for each subject. By contrast, the information-theoretic approaches are computed over the entire window of the fMRI scan (1200 fMRI volumes).

S4.1 Cortical profiles

Figure S8 shows the results of our comparison. We projected the violating triangles, homological scaffold, O-information, and S-information onto the cortical surface (Schaefer 100), as shown in panels **a-f**. We found a moderate correlation (Pearson's correlation coefficient ≈ 0.42) between the topological activations and those from the information-theoretic methods. The correlation values vary depending on whether we consider activation at the nodal level (after projection) or at the original triangle level (panels **g** and **h**).

Furthermore, brain surface projections of the scaffold-to-triangles ratio and the redundancy-to-synergy gradient scores

(based on their respective ranks) also revealed similar cortical patterns ($\rho = 0.41$). Notice that the redundancy and synergy contributions are derived from the O-information vector of size $\binom{N}{3}$ isolating only the positive and negative contributions, respectively. In simpler terms, in resting-state fMRI data, we find that the violating triangle activation pattern resembles redundancy, while the homological scaffold aligns more with synergistic contributions.

Figure S8: **Cortical profiles of topological and information-theoretic measures.** We report the activation of the local higher-order measures, namely (a) violating triangles and (b) homological scaffold, and those identified by the information-theoretic approaches, (d) O-info and (e) S-info, respectively. Brain surface projections of the (c) scaffold-to-triangles ratio and the (f) redundancy-to-synergy gradient scores (based on their respective ranks) also reveal similar patterns, as confirmed by the Pearson’s correlation coefficient of $\rho = 0.42$. We also compare between the topological activations and the corresponding maps obtained from the information-theoretic methods, either at the (g) nodal level (after projection) or at the (h) original triangle level.

The discussion acknowledges the increased computational burden of higher-order approaches. While these methods offer valuable insights, their practical application will be perhaps limited by the computational challenges they pose. Addressing these challenges will be crucial for making higher-order analyses more accessible and feasible in broader research contexts. Could you please provide some quantitative indication of the complexity of calculations required compared to the simpler pair-wise methods?

Reply 1.3: We thank the referee for this comment. The computational complexity of the TDA method is theoretically $O(N^k)$, where k represents the maximum group order considered. In practical terms, computing violating triangles, homological scaffold, and all global higher-order indicators for a single session of HCP data (comprising 119 regions and 1200 time points) takes approximately 5 minutes on a workstation equipped with a Ryzen Threadripper 3970 processor using 24 cores. This computation time is significantly longer with respect to functional connectivity (a few seconds) or edge functional connectivity (10-15 seconds). Potential strategies to mitigate the computation cost include temporal sampling or sampling fewer higher-order interactions (e.g., only a subset of the total number of triangle combinations). While we are currently investigating these approaches, we defer detailed analyses to future studies.

Action taken 1.3: In the last section of the revised main text, we have included a paragraph stating the limitation of our TDA approach in terms of the computational cost associated with computing global and local measures. For completeness, the new part in the main manuscript now reads:

DISCUSSION

...

A key limitation of our current approach is its computational cost. Analyzing co-fluctuation patterns up to order k results in a time complexity of $O(N^k)$, which is significantly higher than methods based solely on node (or edge) connectivity. Future research should explore strategies for reconstructing only a subset of the k -order interactions. This could significantly reduce computational complexity, which currently requires around 5 minutes to process data per HCP subject on a powerful workstation (Ryzen Threadripper 3970 with 24 cores).

COMMENT 1.4

Authors used the PLSC approach for relating brain and behaviour. The more common/generalized multivariate approach is canonical variate analysis. Please elaborate how PLSC differ and why it is preferred here and compare the two methods performance.

Reply 1.4: We thank the referee for pointing this out. Although canonical variate analysis or Canonical Correlation Analysis (CCA) is commonly employed in the literature, its suitability depends on the nature of the data under investigation. In the original manuscript, we considered Partial Least Squares Correlation (PLSC) due to its compatibility with high-dimensional data. PLSC is indeed often favored over CCA for managing high-dimensional data, where the number of variables in one set greatly exceeds that in the other. This preference arises from CCA's sensitivity to scenarios where one set has significantly more variables (p) than the other (q), particularly when $p \gg q$. In contrast, PLSC tends to be less affected by this issue and exhibits greater robustness to noise. This is because CCA aims to maximize the correlation between latent variables from X and Y , PLSC focuses on maximizing covariance.

Nevertheless, we conducted a comparative analysis of both approaches considering the performance of the four metrics considered in our manuscript (i.e., lower- and higher-order methods). In particular, in light of a recent study on the stability of PLSC and CCA with a high number of features [M. Helmer, et al. *Commun Biol* 7, 217 (2024)], we restricted our analysis to a few brain features extracted using the highest Median Absolute Deviation (MAD) value, as done in previous work [C.H. Xia, et al. *Nat Commun* 9, 3003 (2018)] to reduce input features.

Across all methods examined in our study (i.e., FC, edge FC, homological scaffold, and violating triangle), also in this case, both PLSC and CCA consistently showed the superiority of higher-order approaches over lower-order methods (in

terms of covariance explained and canonical correlation, respectively).

Action taken 1.4: In the updated version of the manuscript, we rephrased part of the results to include the additional analysis comparing CCA and PLSC results. Moreover, we included new additional figures in the Supplementary Section S3, providing further evidence of the robustness and reliability of the original PLSC analysis, as well as when considering CCA.

For completeness, we report below the new paragraphs in the main text and in Supplementary Sections. New text is highlighted in red.

RESULTS — BRAIN-BEHAVIOR ASSOCIATION

...

To further validate the robustness of this result, Supplementary Section S3 provides additional analyses of the PLSC approach using a limited set of brain features [70]. We also evaluate the performance of methods using the Canonical Correlation Analysis (CCA) approach [71]. In both cases, at the local level, higher-order methods outperform lower-order approaches for most of the several functional resting-state networks.

SUPPLEMENTARY SECTION S3 — PARTIAL LEAST SQUARE CORRELATION (PLSC) AND CANONICAL CORRELATION ANALYSIS (CCA)

This section delves deeper into the performance of PLSC used in the main text, while also comparing the results against another multivariate method, such as CCA (Canonical Correlation Analysis). We first analyze the total covariance explained by each method using PLSC (same as Figure 4), and the brain-behavior correlation of the significant latent variables.

Figure S5 summarizes the results. Panel (b) shows the correlation values between the significant latent components ($p < 0.05$). Here, higher-order approaches occasionally yield slightly higher correlations, suggesting a potential advantage, while displaying a high amount of variance.

Nevertheless, a very recent study raised concerns about the stability of PLSC and CCA when considering a high number of features per subject [7]. To address this, we performed an additional analysis restricting ourselves to a few brain features extracted using the highest Median Absolute Deviation (MAD) value to reduce the number of features, similar to previous work [10].

Figure S6 displays the results of the PLSC analysis for this reduced feature set (i.e. MAD features = 10). While the total covariance values differ from those reported in the main text, the overall trend remains consistent. As we move from lower-order methods like functional connectivity (FC) or edge functional connectivity (eFC) to higher-order methods like violating triangles or scaffolds, we observe a significant increase in the explained brain-behavior covariance. The only exception is the FP network. When examining the brain behavior correlation, all methods achieve similar performance, with higher-order approaches providing slightly higher mean correlations in some networks.

We now shift our focus to examining the performance of the four approaches using Canonical Correlation Analysis (CCA) to assess brain-behavior associations. For this comparison, we relied on the R code provided in Ref. [10]. Here, the 100 HCP participants are randomly stratified into a discovery ($n = 67$) and a replication sample ($n = 33$).

It is important to note that PLSC is often favored over CCA when dealing with high-dimensional data, as mentioned in Ref. [7]. This preference is due to CCA's sensitivity in situations where one set has significantly more variables (p) than the other (q), particularly when $p \gg q$. In contrast, PLSC tends to be less affected by this issue and demonstrates greater robustness to noise. While CCA aims to maximize the correlation between latent variables from matrices X and Y , PLSC focuses on maximizing covariance, with X representing brain features and Y representing behavioral features.

Given the potential stability issues of CCA with high-dimensional data [7], we again used the same 10 brain features identified using MAD. Figure S7 summarizes the results of our investigation. In particular, Figure S7a reports the canonical correlation induced by the leading significant latent variable. In a similar way as shown for the PLSC covariance analysis, also in this case higher-order approaches typically perform better than lower-order methods, with the only exception of the VA network. Conversely, when focusing on the total covariance explained by the significant latent components

Figure S5: **Behavioral significance across methods for fMRI resting-state data.** (a) Same panel as Figure 4b. Percentage of covariance explained by significant latent components ($p < 0.05$) obtained from PLSC analyses, considering the local functional connections extracted from the different methods and 10 cognitive scores. PLSC components were independently assessed for each method. To obtain a more robust estimate of the covariance explained, we consider a bootstrap procedure randomly sub-sampling 80 subjects from the total of 100 and repeating the PLSC analysis 100 times. (b) Correlation between brain and behavioral scores considering the significant latent variables ($p < 0.05$). Higher-order approaches occasionally yield slightly higher correlations, suggesting a potential advantage. Functional networks: visual (VIS), somatomotor (SM), dorsal attention (DA), ventral attention (VA), limbic (L), frontoparietal (FP), and default mode network (DMN).

($p < 0.05$), the eFC method performs better in three out of the seven functional resting-state networks (namely DA, VA, and FP), while higher-order approaches lead in the remaining four.

Figure S6: **PLSC behavioral significance across methods considering a reduced number of brain features.** (a) Percentage of the total covariance explained by significant components ($p < 0.05$) obtained from PLSC analyses, considering the top 10 functional connections extracted using Median Absolute Deviation (MAD) and 10 cognitive scores. PLSC components were independently assessed for each method. To obtain a more robust estimate of the covariance explained, we consider a bootstrap procedure randomly sub-sampling 80 subjects from the total of 100 and repeating the PLSC analysis 100 times. (b) Correlation score between brain and behavioral scores considering the first significant latent variable ($p < 0.05$). All methods achieve similar performance in this case. Higher-order approaches occasionally yield slightly higher correlations, suggesting a potential advantage.

Figure S7: Behavioral significance across methods using Canonical Correlation Analysis. (a) Canonical correlation associated with the first significant latent variable ($p < 0.05$, highest covariance explained). For each method, this score was obtained by considering CCA between the top 10 functional connections identified using the MAD metric and 10 cognitive scores. Higher-order methods generally display higher correlation scores, except for VA, suggesting a potential advantage. (b) Percentage of covariance explained by the significant latent components ($p < 0.05$) from the CCA analysis. In this case, eFC performed better in three functional networks (DA, VA, and FP), while higher-order approaches excelled in other resting-state functional networks. CCA components were independently assessed for each method. Correlation and covariance estimates were derived by randomly dividing HCP subjects into discovery ($n = 67$) and replication ($n = 33$) sets. Standard errors on the correlation were smaller than 0.01.

Detailed reply to the report of reviewer # 2

COMMENT 2.0

In this study, Santoro et al., use a time-resolved, higher-order analysis (recently introduced by the same authors) to explore whether higher-order interactions provide a significant improvement over classic, pairwise functional connectivity analyses. To do this, they run a battery of tests, including inter-scan subject fingerprinting, task-decoding, and brain-behavior association correlations. They show evidence that the higher-order measures generally outperform the lower-order ones (although with caveats). This is a significant finding for the nascent field of higher-order network neuroscience – many studies have been published using fancy methods to show that higher-order interactions exist, but it’s generally been unclear whether it’s all worth the effort. Santoro et al., show strong evidence that it is worth the effort. I think that the paper is strong and generally well written – I see no glaring issues and I am happy to accept it after minor revisions (generally things that I wish were explained better).

Reply 2.0: We are really grateful to the referee for the positive feedback of our work and for judging the topic of our manuscript of general interest for the field of higher-order network neuroscience. In the revised version of the manuscript, we addressed all the points raised by the reviewer, and we believe that the manuscript now provides new details, analyses, and clarifications of the results presented in the main text.

COMMENT 2.1

*I am struggling to understand how to interpret these higher-order edge time series. I get that for pairs of regions, the ETS is essentially a local decomposition of the Pearson product-moment coefficient (Betz et al., showed this, if memory serves). However, there is no general N-way generalization of the Pearson coefficient, so I’m not entirely sure what the product of $x_1 * x_2 * \dots * x_N$ really means. I would like the authors to spend some time meditating on this – if the construction is just a mathematical “trick” to juice the performance of the methods then that’s fine (statistics is full of such things), but it is a little bit unsatisfying as-is.*

Reply 2.1: We appreciate the referee’s comment and acknowledge the challenge of interpretation. In the original method described in the first paper [A. Santoro, et al. Nat. Phys. 19, 221–229 (2023)], we faced the difficulty of generalizing Pearson’s correlation in a unique manner. Among the different options, we decided to use the product of the signal $x_1 * x_2 * \dots * x_N$ with sign remapping, with the aim of mainly identifying coherent contributions over time, i.e. the coherent group co-fluctuations. While this approach is like a mathematical “trick,” analyzing a specific higher-order time series post z-scoring allows us to determine whether the corresponding group of node signals is coherent at a given time. Moreover, during the development of the original methodology, we explored various alternative formulations for “reconstructing” higher-order signals using linear and non-linear combinations of node signals (e.g., extending and using existent information-theoretic measures to multiple signals). However, most of these alternative formulations failed to differentiate between different spatiotemporal patterns generated by specific dynamical systems, such as Coupled Map Lattices.

Action taken 2.1: In the updated manuscript, we rephrased a portion of the methodology explanation in the methods and introduced a new paragraph to provide further clarity on the rationale behind the choice of the product of the time series.

For reference, the new paragraphs in the main text are provided below. New additions are highlighted in red.

METHODS — CLASSICAL METHODS AND TOPOLOGICAL HIGHER-ORDER ORGANIZATION OF FMRI SIGNALS

As done in Ref. [41], also in this work we only consider co-fluctuations of dimension up to $k = 2$, so that triangles represent the only higher-order structures in the weighted simplicial complex \mathcal{K} , and weights on the simplices, i.e. w_{ij} and w_{ijk} , represent the magnitude of edges and triangles co-fluctuations. **We note that while the concept of edge time**

series represents a temporal unwrapping of the Pearson's correlation coefficient [51], the generalization to k -order co-fluctuations does not correspond to any specific statistical measure. However, the sign remapping emphasizes the purely coherent co-fluctuations, allowing us to better focus and discriminate these contributions.

COMMENT 2.2

Naively, shouldn't the parity be based on sum of signs modulo 2? For example, $-1 \times -1 \times 1$ has parity of 1, but in your schema it would be assigned a negative sign? Am I understanding that right? The only moments that get a positive score are either all negative or all positive? The product of the z -scores is already a signed value – why change it?

Reply 2.2: The sign scheme we introduced is rooted in our emphasis on coherence. Under this scheme, group co-fluctuations are mapped as positive only if all nodal signals are either consistently negative or positive, indicating perfect coherence. Conversely, all other cases result in negative mappings, reflecting discordant signs. Thus, when examining positive triangles, especially the violating ones, we can effectively pinpoint the most significant instances of coherent group fluctuations within the brain network, knowing beforehand that they originate from perfectly coherent nodal signals. Moreover, while the product of z -scores inherently carries a sign (positive or negative), the product does not allow to directly distinguish between coherent and incoherent fluctuations, which is the core objective of our approach. The sign remapping step is therefore crucial for making this distinction.

Action taken 2.2: In the revised version of the manuscript, we clarified the formulation of the sign scheme to explicitly mention the emphasis on perfect coherence.

For completeness, the new part in the main manuscript now reads (changes are highlighted in red):

RESULTS

We finally assign a sign to the resulting k -order time series at each time based on a strict parity rule: positive for fully concordant group interactions (nodes times series have all positive or all negative values at that timepoint), and negative for discordant interactions (a mixture of positive and negative values). **Notice that this sign remapping allows us to explicitly focus on perfectly coherent contributions, which are always marked as positive (see also Methods for the analytical formulation).**

METHODS - CLASSICAL METHODS AND TOPOLOGICAL HIGHER-ORDER ORGANIZATION OF fMRI SIGNALS

In order to differentiate concordant group interactions from discordant ones in a k -order product, concordant signs are always positively mapped, while discordant signs are negatively mapped. That is,

$$\text{sign} [\xi_{0\dots k}(t)] := (-1)^{\text{sgn}[(k+1)-|\sum_0^k \text{sgn}[z_i(t)]|]},$$

where $\text{sgn}[\bullet]$ is the signum function of a real number. Formally, the weight $w_{0\dots k}(t)$ of the k -order co-fluctuations at time t is defined as:

$$w_{0\dots k}(t) = \text{sign}[\xi_{0\dots k}(t)]|\xi_{0\dots k}(t)|$$

In this way, all the perfectly coherent contributions (namely, either all positive or all negative) will be mapped as positive. This is because in our method we are mainly focusing on the coherent contributions within the multivariate time series.

COMMENT 2.3

Have you considered other ways of tracking the interactions of the elements? For example, for each TR, the instantaneous z -scores define a vector: perhaps you could use the Manhattan norm, Euclidean norm, Chebyshev norm, etc? Since a k -dimensional time series is a trajectory through k -d space, taking the norm at each point seems more interpretable to me.

Reply 2.3: We appreciate the referee's insightful observation, as these approaches are indeed valuable for tracking interactions. During the development of our initial work, we explored various methods for tracking interactions, such as

spectral decomposition and geometrical embedding. However, in certain synthetic benchmarks generated by the Coupled Map Lattice, these alternative approaches exhibited poor performance. We are currently investigating the possibility of modifying our approach by relying on Takens' embeddings to embed the trajectories of these interactions into metric spaces. While we have some initial results in controlled settings (e.g., simulated time series), we believe a more in-depth theoretical exploration is needed before presenting these findings. We plan to include these results in future (theoretical) studies.

Action taken 2.3: In the revised version of the manuscript, we appended a paragraph at the conclusion of the main text, outlining potential future directions for the method.

The new paragraph is included below, with the additional text highlighted in red for your reference:

DISCUSSION

...

Furthermore, integrating techniques based on Takens' embeddings, spectral decomposition, or other geometric approaches might offer valuable insights into the causal relationships between brain activity patterns at consecutive time points. These approaches could potentially complement our current method and provide a more comprehensive understanding of brain network dynamics.

COMMENT 2.4

*In the same vein: if you have k time series and use element-wise multiplication to compress them down to a 1-dimensional time series, haven't you *lost* a lot of information about the interaction? The move from a k -dimensional space to a 1-dimensional space is necessarily degenerate: there are many sets of k numbers that will produce the same product, so you've lost a lot of information in the joint-state of the k time series (which, to my mind, seems like it would be important for a higher-order analysis). Is this an analysis of higher-order redundancy or higher order synergy?*

Reply 2.4: We thank the referee for this comment. It is true that a 1-dimensional time series of group k interaction can stem from various combinations of k time series, leading to a loss of information about the original interaction. However, this is an inherent challenge in the reconstruction problem, which is inherently ill defined. Nonetheless, two crucial points need highlighting.

First, our emphasis on coherent interactions mitigates information loss, particularly in the context of positive interactions, as the number of potential combinations leading to positive contributions decreases with increasing group order. Second, considering all possible k -order interactions within the framework of TDA ensures that we retain complete information about the original signals. For instance, when analyzing 4 time series and examining edges and triplets, the weighted simplicial complex encompasses all $\binom{4}{3}$ triplets and $\binom{4}{2}$ edges, preserving the entirety of information. As a matter of fact, the concept of "information loss" is also present in other common methods, such as ETS proposed by Betzel et al., or classical functional connectivity (FC). In ETS, information loss occurs during local decomposition of the Pearson product-moment coefficient for a pair of time series, while in FC, averaging across time results in information loss. However, when considering the entirety of all possible pairs it allows to retain information about the original signals.

Finally, to further delineate the distinction between redundancy/synergy approaches and our TDA method, we conducted a comparison on static resting-state fMRI data. Specifically, we projected violating triangles and the homological scaffold onto the cortical surface, revealing a moderate correlation (Pearson's correlation $\rho \approx 0.4$) with the corresponding maps obtained from redundancy and synergy analyses, respectively.

Action taken 2.4: In the updated version of the manuscript, we amended part of the introduction, highlighting the ill-defined problem of reconstruction, with the inherent information loss. We additionally included new analyses in Supplementary Section S4.1 comparing the synergy/redundancy measure against our TDA approach in the main text.

The new text, highlighted in red now reads:

INTRODUCTION

This approach builds upon prior work involving edge-level signals and the extension of functional connectivity research beyond pairs [51,52], leveraging low-order signals to define higher-order ones (a reconstruction task that is in general an ill-posed inverse problem).

SUPPLEMENTARY SECTION S4 — COMPARISON WITH MULTIVARIATE INFORMATION-THEORETIC APPROACHES

In this section, we compare the topological approach presented in the main text with recent static higher-order information-theoretic approaches based on O-information, S-information, and Dual Total Correlation (DTC) [8]. We relied on the code provided in Ref. [6] for this comparison.

We evaluated the similarity between the activation patterns of local higher-order measures (violating triangles and the homological scaffold) identified by our topological approach and those identified by the information-theoretic approaches. We remark a crucial difference between the methods: Our topological approach inherently captures temporal dynamics, therefore we need to average the activation patterns across time for each subject. By contrast, the information-theoretic approaches are computed over the entire window of the fMRI scan (1200 fMRI volumes).

S4.1 Cortical profiles

Figure S8 shows the results of our comparison. We projected the violating triangles, homological scaffold, O-information, and S-information onto the cortical surface (Schaefer 100), as shown in panels **a-f**. We found a moderate correlation (Pearson's correlation coefficient ≈ 0.42) between the topological activations and those from the information-theoretic methods. The correlation values vary depending on whether we consider activation at the nodal level (after projection) or at the original triangle level (panels **g** and **h**).

Furthermore, brain surface projections of the scaffold-to-triangles ratio and the redundancy-to-synergy gradient scores (based on their respective ranks) also revealed similar cortical patterns ($\rho = 0.41$). Notice that the redundancy and synergy contributions are derived from the O-information vector of size $\binom{N}{3}$ isolating only the positive and negative contributions, respectively. In simpler terms, in resting-state fMRI data, we find that the violating triangle activation pattern resembles redundancy, while the homological scaffold aligns more with synergistic contributions.

Figure S8: Cortical profiles of topological and information-theoretic measures. We report the activation of the local higher-order measures, namely **(a)** violating triangles and **(b)** homological scaffold, and those identified by the information-theoretic approaches, **(d)** O-info and **(e)** S-info, respectively. Brain surface projections of the **(c)** scaffold-to-triangles ratio and the **(f)** redundancy-to-synergy gradient scores (based on their respective ranks) also reveal similar patterns, as confirmed by the Pearson’s correlation coefficient of $\rho = 0.42$. We also compare between the topological activations and the corresponding maps obtained from the information-theoretic methods, either at the **(g)** nodal level (after projection) or at the **(h)** original triangle level.

COMMENT 2.5

Why do ETS perform so much worse than either FC or HOI? Since the ETS is basically an “unrolling” of the pairwise correlation, it seems like everything available in the FC should also be available in the ETS?

Reply 2.5: In various applications, ETS demonstrates diverse performance outcomes, but it does not consistently rank as the least effective method. For example, in task decoding, ETS outperforms FC and shows similar efficacy to triangles in identifying task/rest blocks, as measured by Element-Centric Similarity (ECS). While ETS may lag in brain fingerprint-

ing, especially when focusing on the 7 resting-state networks, it remains somewhat competitive in certain networks like Visual. Additionally, in the context of whole-brain connections, prior research (Y. Jo, et al. *NeuroImage*, 238:118204, 2021) suggests that FC and eFC display comparable performance, particularly with fewer than 1000-1200 time points, while eFC surpasses FC for longer scans. In PLSC analyses for brain-behavior association, ETS/eFC generally outperforms FC and scaffold, often ranking only second to the violating triangle metric in specific scenarios.

Overall, while the underlying information is consistent across methods, filtering procedures (e.g., considering only a subset of all possible triangles, such as violating ones) or temporal unwrapping of the Pearson's coefficient (i.e. the ETS) may "clean" the signals, leading to better performance in specific applications and/or conditions.

Action taken 2.5: —

COMMENT 2.6

I'm not sure what this sentence means:

"While recent technologies now allow us to record higher-order phenomena at the micro-scale neuronal level (e.g., the simultaneous firing of groups of neurons) [35–37], such data remains unavailable to explore large-scale brain dynamics."

What does "unavailable" mean? Not shared? Or we don't have the tools to get at the HOIs?

Reply 2.6: We appreciate the reviewer's comment. The original text aimed to highlight that in humans, non-invasive techniques like EEG or fMRI cannot directly measure "true" higher-order brain activity. These techniques can only provide noisy estimates of the underlying neural signals. However, advancements in technology offer new possibilities. For instance, Neuropixel probes allow researchers to record the simultaneous firing of neuronal groups in mice. Therefore, this technology provides a closer look at the 'ground truth' of higher-order neural activation patterns.

Action taken 2.6: In the revised version of the manuscript, we have amended the part of the introduction, which now reads (new text highlighted in red):

INTRODUCTION

...

Recent advancements in technology have allowed us to record higher-order phenomena at the micro-scale neuronal level, such as the simultaneous firing of groups of neurons in mice [35-37] or monkeys [38]. However, this type of data is not yet available in humans for studying large-scale brain dynamics. Instead, we typically rely on non-invasive techniques like M/EEG and fMRI, which provide only noisy estimates of the neural activity and cannot directly capture "true" higher-order brain functions. Thus, researchers must rely on statistical methods to infer higher-order interactions from neuroimaging signals recorded from regions of interest. ...

COMMENT 2.7

Small stylistic thing: the transition into "While recent edge-centric..." is a bit jarring.

Reply 2.7: Thanks for pointing this out, we have amended that paragraph.

Action taken 2.7: We have rephrased that part, which now reads (new text highlighted in red):

INTRODUCTION

... Furthermore, early examples of inferred temporal HOI statistics [41] have been successfully used as features in machine learning classifiers to detect financial crises and classify disease types based on their spatial spreading patterns,

providing better accuracy when compared to measures based on pairwise descriptions (lower order), such as edge-centric approaches [51,52]. Currently, edge-centric approaches in fMRI have shown promise in identifying overlapping brain communities and estimating dynamic connectivity at finer timescales [53] compared to classical functional connectivity (FC) methods. In contrast, the potential benefits of HOIs in fMRI data analysis have received limited exploration. Specifically, it remains uncertain whether reconstructed HOIs offer advantages and deeper insights over conventional methods for fMRI data analysis.

In this study, we address this question by leveraging a recent topological approach capable of reconstructing HOI structures at the temporal level [41]. Our analysis focuses on resting state and tasks fMRI data from 100 unrelated subjects of the Human Connectome Project (HCP).

...

Detailed reply to the report of reviewer # 3

COMMENT 3.0

This is an interesting and generally well-written paper. The report brings together number of technical advancements in fMRI. Ultimately however, it often feels the “whole is less than the sum of its parts”. While it is exciting to see recent extensions of edge time series further extended, it remains a bit unclear how these methods compare to other for example high dimensional functional data analysis techniques (e.g., ICA) for the reviewed use cases or what the main take aways are from these different analyses that often lead to mixed evidence. While this alone isn’t a problem writ large with the analyses here, a lack of clarity on alternative high dimensional functional data techniques here for certain analyses (e.g., prediction of phenotypes, fingerprinting/differential heritability) makes it difficult to determine whether just more complex representations of data are outperforming less complex ones, or whether this particular framework has additional purchase. Additional specific comments are listed below:

Reply 3.0: We appreciate the referee’s positive evaluation and interest in our work. It is important to clarify that Independent Component Analysis (ICA) serves a different purpose than the connectivity approaches discussed in our manuscript. To our knowledge, ICA is used for dimensionality reduction, filtering, or pre-processing of fMRI data, aiming to extract either independent temporal or spatial components. Comparing the results of brain fingerprinting and the PLSC approach (both at the global and local level) from our methods with those obtained using ICA is not entirely straightforward. Additionally, determining the number of ICA components and how to match components across multiple ICA runs remains a topic of ongoing debate in the literature.

Nevertheless, in our revised manuscript, we utilize a recent robust ICA pipeline on connectomes (i.e. connICA [E. Amico et al, NeuroImage, 148:201–211, 2017]) to thoroughly illustrate the advantages of higher-order approaches over ICA, particularly in the context of brain fingerprinting and brain-behavior association tasks. As a result, we now extensively demonstrate the additional advantages of taking into account three-body interactions over lower-order representations. A detailed point-to-point response is reported below.

COMMENT 3.1

Theme 1. Framing and level of complexity.

A. Its is difficult to determine for the reader in the intro the throughline the connects the conducted analyses. As stated above, an extension of edge time series seems interesting but most of the included analyses focus on use cases outside of network neuroscience or functional neuroanatomy, but rather prediction (behavior, task decoding). This is okay in theory, but the general premise can be a bit hard to follow. Are the authors suggesting this should be how new investigators analyzed fMRI data in prediction contexts? Much more context would be helpful if that were the case and if not to better clarify what exact value add these analyses have?

Reply 3.1: We thank the referee for this observation. Our manuscript investigates whether higher-order connectivity methods improve predictions (behavior, task decoding) and fingerprinting compared to traditional approaches, like functional connectivity (FC) or lower-order methods like edge FC (eFC). We found that for whole-brain analyses, all methods performed similarly. Therefore, higher-order approaches may not be necessary in these cases.

However, when analyzing data at the level of the 7 resting-state networks, results suggest that higher-order approaches might provide additional information. Therefore, we do not recommend using these approaches universally for all fMRI data analyses. Instead, we suggest employing higher-order methods for local-level analyses when higher performance is desired, while keeping in mind the increased computational time and the more complex interpretation of the results.

Action taken 3.1: In the revised version of the manuscript, we have rephrased part of the introduction and discussion to better highlight our contribution and distinguish it from ICA approaches.

The new paragraphs are reported below, with the additional text highlighted in red for your reference:

INTRODUCTION

...

These emerging approaches, applicable to both healthy [45] and clinical populations [46,47], therefore represent a fundamental shift from methods like motif analysis or classical approaches like ICA [48,49]. Initial applications of information-theoretic techniques in fMRI suggested that higher-order dependencies reconstructed from fMRI data can encode meaningful brain biomarkers, including the ability to differentiate patients in different states of consciousness [20,23] or detect effects associated with age [50]. Furthermore, early examples of inferred temporal HOI statistics [41] have been successfully used as features in machine learning classifiers to detect financial crises and classify disease types based on their spatial spreading patterns, providing better accuracy when compared to measures based on pairwise descriptions (lower order), such as edge-centric approaches [51,52]. Currently, edge-centric approaches in fMRI have shown promise in identifying overlapping brain communities and estimating dynamic connectivity at finer timescales [53] compared to classical functional connectivity (FC) methods. In contrast, the potential benefits of HOIs in fMRI data analysis have received limited exploration. Specifically, it remains uncertain whether reconstructed HOIs offer advantages and deeper insights over conventional methods for fMRI data analysis.

DISCUSSION

...

Our findings outline the substantial improvements that these higher-order methods could bring to the neuroscientific community: here, enabling superior task differentiation, refining functional brain fingerprinting, and establishing more robust associations between brain activity and behavior. This contrast highlights the limitations of relying solely on traditional pairwise or bivariate methods, and the fundamental differences from other computational methods like ICA [48,66]. It advocates for a more nuanced approach, urging researchers to confront the computational challenges posed by higher-order interactions. The true strength of these methods lies in their ability to decipher subtle, localized brain activity patterns that might otherwise remain obscured when using conventional techniques. This emphasizes the significance of embracing advanced methodologies to fully unlock the potential of fMRI data analysis, advancing the understanding of the intricate landscape of brain function.

COMMENT 3.2

B. The introduction is written generally very well for a broad audience. That said, connecting the dots here and keeping the high-level concepts clear would be important for a broad audience at this outlet.

Reply 3.2: We appreciate the referee's feedback on the introduction for being well-written for a general audience. However, their specific critique regarding connecting the dots and maintaining clarity for high-level concepts is not entirely clear to us. Therefore, we revised part of the introduction to better clarify the goal of the manuscript and distinguish our contribution from other methods like ICA or PCA. This should prevent any potential confusion for the reader.

Action taken 3.2: We rephrased part of the introduction to improve clarity on the goal of our work and to state the difference between our approach and other classical fMRI methods.

For completeness, the new part in the main manuscript now reads (new text highlighted in red, but same as in comment 3.1):

INTRODUCTION

...

These emerging approaches, applicable to both healthy [45] and clinical populations [46,47], therefore represent a fundamental shift from methods like motif analysis or classical approaches like ICA [48,49]. Initial applications of

information-theoretic techniques in fMRI suggested that higher-order dependencies reconstructed from fMRI data can encode meaningful brain biomarkers, including the ability to differentiate patients in different states of consciousness [20,23] or detect effects associated with age [50].

...

Currently, edge-centric approaches in fMRI have shown promise in identifying overlapping brain communities and estimating dynamic connectivity at finer timescales [53] compared to classical functional connectivity (FC) methods. In contrast, the potential benefits of HOIs in fMRI data analysis have received limited exploration. Specifically, it remains uncertain whether reconstructed HOIs offer advantages and deeper insights over conventional methods for fMRI data analysis.

COMMENT 3.3

Theme 2: Lack of comparisons to other functional data analysis techniques given the lack of clarity on the goal (see above). This likewise extends to the methods used for prediction of phenotypes and decoding.

Reply 3.3: We thank the referee for this comment. In the original manuscript, we compared higher-order approaches (scaffold and triangles) against classical functional connectivity and edge functional connectivity. Since ICA (as well as PCA or other methods) generally differ from connectivity approaches, they were not included initially. However, in the revised manuscript, we have applied the connICA framework [E. Amico et al, NeuroImage, 148:201–211, 2017] to functional connectomes to provide an additional comparison using ICA in terms of brain-behavior association and brain fingerprinting. In the context of brain fingerprinting, we demonstrate that connICA does not perform well when compared with higher-order approaches.

Action taken 3.3: We provide an additional analysis in Supplementary Section S5 to compare ConnICA against the approaches considered in the main text. We show that ConnICA has a lower performance in the context of brain fingerprinting.

For completeness, the new part in the main manuscript and Supplementary Material now reads (new text highlighted in red):

RESULTS — FUNCTIONAL BRAIN FINGERPRINTING

...

Comparisons with other methods, including static information-theoretic approaches [40,50] and variants of ICA [66], are provided in the Supplementary Sections S4 and S5, respectively. Although static information-theoretic approaches show a slight improvement over our temporal topological method when analyzing the entire fMRI scan at once, ICA exhibits lower performance.

SUPPLEMENTARY SECTION S5 — CONNICA COMPARISON

This section compares the methods presented in the main text with the ConnICA approach [4,5]. ConnICA utilizes Independent Component Analysis (ICA) to extract stable independent functional connectivity patterns (FC-traits) from a set of individual functional connectomes, without imposing any *a priori* data stratification into groups. The original ConnICA study investigated the links between FC-traits derived from resting-state fMRI and cognitive/clinical features related to consciousness levels. Here, we apply ConnICA with the goal of identifying stable FC traits that can be used for brain fingerprinting. Similarly to what was done in Ref. [4], we set the number of ICA components to 20 and independently applied ConnICA to the resting-state fMRI data of 100 HCP subjects across two sessions (day 1 and day 2). Figure shows the results of ConnICA approach for brain fingerprinting. First, only one FC-trait exhibited a moderate correlation (Pearson's correlation coefficient $\rho \approx 0.5$) between the two sessions. All other traits showed low similarity (see panel **a**). We rely on the most robust FC trait between the sessions to quantify whole-brain fingerprinting accuracy. Specifically, we consider Intraclass Correlation Coefficient (ICC) [9] to assess the paired FC-trait weights across subjects

(see panel **b**). We remind that ICC is a statistical measure used to determine the agreement between ratings/scores of different groups. In other words, the stronger the agreement between two or more groups, the higher the ICC value. Finally, we reconstruct functional connectomes using only FC-traits with a correlation coefficient above 0.4 for both sessions. This allows us to perform similar analyses to those presented in Figure 3 of the main text. As shown in Figure **c**, higher-order approaches outperform ConnICA, which exhibits quite a high variance across multiple bootstrapped runs (sub-sampling only 80 out of 100 HCP subjects at each of the 100 iterations).

Figure S11: **ConnICA approach for brain fingerprinting.** (a) Pairwise Pearson’s correlation between all the robust FC-traits independently identified using ConnICA for the two HCP sessions. Only one FC-trait showed a correlation slightly above 0.5, indicating low similarity of the robust FC traits between the sessions. (b) Comparing the weights of the robust FC-trait associated with each subject in the two different sessions yields a low Intra-Class Correlation (ICC). This reflects the difficulty of accurately identifying subjects based on FC-traits alone. (c) We reconstructed functional connectomes using only FC-traits with a correlation coefficient greater than 0.4 for both sessions. The differential identifiability scores for the ConnICA approach, calculated at the level of the seven resting-state networks, confirm its lower performance compared to the higher-order methods presented in the main text. Additionally, ConnICA exhibits higher variability across multiple bootstrapped runs. Results in panel **b-c** are obtained using only 80 out of the 100 HCP subjects, for 100 independent iterations.

COMMENT 3.4

Theme 3: Magnitude/scaling of brain-behavior phenomena.

A. It is a bit unclear as to what exact scaling is happening in Figure 4 but these values are very very high. Higher than

predictions from multiple independent groups for these same types of behaviors (e.g. Marek et al., Nature, 2022; Chen et al., Nature Communications, 2022; Schulz et al., Cell Reports, 2024) each of which consider multiple methods. Can the authors clarify why this is?

Reply 3.4: We thank the referee for this important point. We have investigated this issue and want to clarify the following points:

- **Figure 4:** The figure shows the total covariance explained by different brain features for various methods. This differs from predictions based on linear models, as reported by Marek et al. (Nature, 2022), Chen et al. (Nature Communications, 2022), and Schulz et al. (Cell Reports, 2024), which report correlation values. Techniques like Partial Least Squares Correlation (PLSC) and Canonical Correlation Analysis (CCA) allow for the assessment of brain-behavior correlations in a multivariate manner. Since both PLSC and CCA rely on a SVD decomposition of the covariance matrix obtained from brain and behavioral features, we can quantify the amount of total covariance explained by the identified latent components, as well as their correlation with respect to brain/behavior. Typically, correlations result in low values in large populations, while covariance can reach higher values. In the revised version, we also report in Supplementary Section S3 the correlations to show the differences between methods and, as the referee pointed out, the values are lower;
- **Recent Critique:** Similar claims to those presented by Marek et al. can be obtained using different statistical approaches, as evidenced by recent critiques [T. Spisak, et al. Nature 615, E4–E7 (2023)];
- **New Correlation Results:** We have performed new analyses using PLSC and CCA. The new correlation results from the PLSC analysis, as well as those obtained using CCA, consistently show the improvement of higher-order approaches over lower-order methods. The correlation values in these analyses are much lower (around 0.4 for Pearson’s correlation) than the covariance values reported in the original text. Based on a recent study on the stability of PLSC and CCA with a high number of features [M. Helmer, et al. Commun Biol 7, 217 (2024)], we have also restricted some of the new analyses to a few brain features extracted using the highest Median Absolute Deviation (MAD) value, as done in previous work [C.H. Xia, et al. Nat Commun 9, 3003 (2018)] to reduce input features. Also in this case, across all methods examined in our study (i.e., FC, edge FC, homological scaffold, and violating triangle), both PLSC and CCA consistently demonstrated the improvement provided by higher-order approaches over lower-order methods.
- **Mistake in panel 4b-c for FC and eFC:** We identified a bug in the code for selecting the corresponding connections of interest for FC and eFC. Also, the panel c now include only stable cognitive saliences over bootstrap realizations.

Finally, to ensure reproducibility, we will release our Python/Julia/R/Matlab code to reproduce all the analyses and figures in a public repository.

Action taken 3.4: In the revised version of the manuscript, we have modified and clarified the presentation of the results of Figure 4. Additionally, we have included new supplementary figures in Section S3 that show the results of the Canonical Correlation analyses and the correlations between the latent components identified using the Partial Least Squares Correlation approach.

For completeness, we report below the new paragraphs in the main text and in Supplementary Sections. New text is highlighted in red.

RESULTS — BRAIN-BEHAVIOR ASSOCIATION

...

To further validate the robustness of this result, Supplementary Section S3 provides additional analyses of the PLSC approach using a limited set of brain features [70]. We also evaluate the performance of methods using the Canonical

Correlation Analysis (CCA) approach [71]. In both cases, at the local level, higher-order methods outperform lower-order approaches for most of the several functional resting-state networks.

SUPPLEMENTARY SECTION S3 — PARTIAL LEAST SQUARE CORRELATION (PLSC) AND CANONICAL CORRELATION ANALYSIS (CCA)

This section delves deeper into the performance of PLSC used in the main text, while also comparing the results against another multivariate method, such as CCA (Canonical Correlation Analysis). We first analyze the total covariance explained by each method using PLSC (same as Figure 4), and the brain-behavior correlation of the significant latent variables.

Figure S5 summarizes the results. Panel (b) shows the correlation values between the significant latent components ($p < 0.05$). Here, higher-order approaches occasionally yield slightly higher correlations, suggesting a potential advantage, while displaying high amount of variance.

Nevertheless, a very recent study raised concerns about the stability of PLSC and CCA when considering a high number

Figure S5: **Behavioral significance across methods for fMRI resting-state data.** (a) Same panel as Figure 4b. Percentage of covariance explained by significant latent components ($p < 0.05$) obtained from PLSC analyses, considering the local functional connections extracted from the different methods and 10 cognitive scores. PLSC components were independently assessed for each method. To obtain a more robust estimate of the covariance explained, we consider a bootstrap procedure randomly sub-sampling 80 subjects from the total of 100 and repeating the PLSC analysis 100 times. (b) Correlation between brain and behavioral scores considering the significant latent variables ($p < 0.05$). Higher-order approaches occasionally yield slightly higher correlations, suggesting a potential advantage. Functional networks: visual (VIS), somatomotor (SM), dorsal attention (DA), ventral attention (VA), limbic (L), frontoparietal (FP), and default mode network (DMN).

of features per subject [7]. To address this, we performed an additional analysis restricting ourselves to a few brain features extracted using the highest Median Absolute Deviation (MAD) value to reduce the number of features, similar

to previous work [10].

Figure S6 displays the results of the PLSC analysis for this reduced feature set (i.e. MAD features = 10). While the total covariance values differ from those reported in the main text, the overall trend remains consistent. As we move from lower-order methods like functional connectivity (FC) or edge functional connectivity (eFC) to higher-order methods like violating triangles or scaffolds, we observe a significant increase in the explained brain-behavior covariance. The only exception is the FP network. When examining the brain behavior correlation, all methods achieve similar performance, with higher-order approaches providing slightly higher mean correlations in some networks.

Figure S6: **PLSC behavioral significance across methods considering a reduced number of brain features.** (a) Percentage of the total covariance explained by significant components ($p < 0.05$) obtained from PLSC analyses, considering the top 10 functional connections extracted using Median Absolute Deviation (MAD) and 10 cognitive scores. PLSC components were independently assessed for each method. To obtain a more robust estimate of the covariance explained, we consider a bootstrap procedure randomly sub-sampling 80 subjects from the total of 100 and repeating the PLSC analysis 100 times. (b) Correlation score between brain and behavioral scores considering the first significant latent variable ($p < 0.05$). All methods achieve similar performance in this case. Higher-order approaches occasionally yield slightly higher correlations, suggesting a potential advantage.

We now shift our focus to examining the performance of the four approaches using Canonical Correlation Analysis (CCA) to assess brain-behavior associations. For this comparison, we relied on the R code provided in Ref. [10]. Here, the 100 HCP participants are randomly stratified into a discovery ($n = 67$) and a replication sample ($n = 33$).

It is important to note that PLSC is often favored over CCA when dealing with high-dimensional data, as mentioned in Ref. [7]. This preference is due to CCA's sensitivity in situations where one set has significantly more variables (p) than the other (q), particularly when $p \gg q$. In contrast, PLSC tends to be less affected by this issue and demonstrates greater

robustness to noise. While CCA aims to maximize the correlation between latent variables from matrices X and Y , PLSC focuses on maximizing covariance, with X representing brain features and Y representing behavioral features. Given the potential stability issues of CCA with high-dimensional data [7], we again used the same 10 brain features identified using MAD. Figure S7 summarizes the results of our investigation. In particular, Figure S7a reports the canonical correlation induced by the leading significant latent variable. In a similar way as shown for the PLSC covariance analysis, also in this case higher-order approaches typically perform better than lower-order methods, with the only exception of the VA network. Conversely, when focusing on the total covariance explained by the significant latent components ($p < 0.05$), the eFC method performs better in three out of the seven functional resting-state networks (namely DA, VA, and FP), while higher-order approaches lead in the remaining four.

Figure S7: **Behavioral significance across methods using Canonical Correlation Analysis.** (a) Canonical correlation associated with the first significant latent variable ($p < 0.05$, highest covariance explained). For each method, this score was obtained by considering CCA between the top 10 functional connections identified using the MAD metric and 10 cognitive scores. Higher-order methods generally display higher correlation scores, except for VA, suggesting a potential advantage. (b) Percentage of covariance explained by the significant latent components ($p < 0.05$) from the CCA analysis. In this case, eFC performed better in three functional networks (DA, VA, and FP), while higher-order approaches excelled in other resting-state functional networks. CCA components were independently assessed for each method. Correlation and covariance estimates were derived by randomly dividing HCP subjects into discovery ($n = 67$) and replication ($n = 33$) sets. Standard errors on the correlation were smaller than 0.01.

COMMENT 3.5

B. Also the models perform worse when including more brain regions which likely suggests poorly condition whole brain models.

Reply 3.5: We appreciate the feedback from the referee, although we find it somewhat unclear. Overall, our findings indicate that there are no noticeable differences between approaches when analyzing whole-brain connections. However, when we adopt a local approach, focusing on individual functional resting-state networks, differences become noticeable. Furthermore, in the updated manuscript, we rely on PLSC and CCA to investigate brain-behavior associations. We explore scenarios where the number of brain features is small (brain features = 10) and consistent across different methods, again demonstrating the improvement of the higher-order approaches.

Action taken 3.5: We included new supplementary figures in Supplementary Section S3 to compare the results of the PLSC approach against CCA. For completeness, the new part in the Supplementary Section now reads (changes are highlighted in red, same as comment 3.4):

SUPPLEMENTARY SECTION S3 — PARTIAL LEAST SQUARE CORRELATION (PLSC) AND CANONICAL CORRELATION ANALYSIS (CCA)
This section delves deeper into the performance of PLSC used in the main text, while also comparing the results against another multivariate method, such as CCA (Canonical Correlation Analysis). We first analyze the total covariance explained by each method using PLSC (same as Figure 4), and the brain-behavior correlation of the significant latent variables.

Figure S5 summarizes the results. Panel (b) shows the correlation values between the significant latent components ($p < 0.05$). Here, higher-order approaches occasionally yield slightly higher correlations, suggesting a potential advantage, while displaying high amount of variance.

Nevertheless, a very recent study raised concerns about the stability of PLSC and CCA when considering a high number of features per subject [7]. To address this, we performed an additional analysis restricting ourselves to a few brain features extracted using the highest Median Absolute Deviation (MAD) value to reduce the number of features, similar to previous work [10].

Figure S6 displays the results of the PLSC analysis for this reduced feature set (i.e. MAD features = 10). While the total covariance values differ from those reported in the main text, the overall trend remains consistent. As we move from lower-order methods like functional connectivity (FC) or edge functional connectivity (eFC) to higher-order methods like violating triangles or scaffolds, we observe a significant increase in the explained brain-behavior covariance. The only exception is the FP network. When examining the brain behavior correlation, all methods achieve similar performance, with higher-order approaches providing slightly higher mean correlations in some networks.

We now shift our focus to examining the performance of the four approaches using Canonical Correlation Analysis (CCA) to assess brain-behavior associations. For this comparison, we relied on the R code provided in Ref. [10]. Here, the 100 HCP participants are randomly stratified into a discovery ($n = 67$) and a replication sample ($n = 33$).

It is important to note that PLSC is often favored over CCA when dealing with high-dimensional data, as mentioned in Ref. [7]. This preference is due to CCA's sensitivity in situations where one set has significantly more variables (p) than the other (q), particularly when $p \gg q$. In contrast, PLSC tends to be less affected by this issue and demonstrates greater robustness to noise. While CCA aims to maximize the correlation between latent variables from matrices X and Y , PLSC focuses on maximizing covariance, with X representing brain features and Y representing behavioral features.

Given the potential stability issues of CCA with high-dimensional data [7], we again used the same 10 brain features identified using MAD. Figure S7 summarizes the results of our investigation. In particular, Figure S7a reports the canonical correlation induced by the leading significant latent variable. In a similar way as shown for the PLSC covariance analysis, also in this case higher-order approaches typically perform better than lower-order methods, with the only exception of the VA network. Conversely, when focusing on the total covariance explained by the significant latent components

Figure S5: **Behavioral significance across methods for fMRI resting-state data.** (a) Same panel as Figure 4b. Percentage of covariance explained by significant latent components ($p < 0.05$) obtained from PLSC analyses, considering the local functional connections extracted from the different methods and 10 cognitive scores. PLSC components were independently assessed for each method. To obtain a more robust estimate of the covariance explained, we consider a bootstrap procedure randomly sub-sampling 80 subjects from the total of 100 and repeating the PLSC analysis 100 times. (b) Correlation between brain and behavioral scores considering the significant latent variables ($p < 0.05$). Higher-order approaches occasionally yield slightly higher correlations, suggesting a potential advantage. Functional networks: visual (VIS), somatomotor (SM), dorsal attention (DA), ventral attention (VA), limbic (L), frontoparietal (FP), and default mode network (DMN).

($p < 0.05$), the eFC method performs better in three out of the seven functional resting-state networks (namely DA, VA, and FP), while higher-order approaches lead in the remaining four.

Figure S6: **PLSC behavioral significance across methods considering a reduced number of brain features.** (a) Percentage of the total covariance explained by significant components ($p < 0.05$) obtained from PLSC analyses, considering the top 10 functional connections extracted using Median Absolute Deviation (MAD) and 10 cognitive scores. PLSC components were independently assessed for each method. To obtain a more robust estimate of the covariance explained, we consider a bootstrap procedure randomly sub-sampling 80 subjects from the total of 100 and repeating the PLSC analysis 100 times. (b) Correlation score between brain and behavioral scores considering the first significant latent variable ($p < 0.05$). All methods achieve similar performance in this case. Higher-order approaches occasionally yield slightly higher correlations, suggesting a potential advantage.

Figure S7: Behavioral significance across methods using Canonical Correlation Analysis. (a) Canonical correlation associated with the first significant latent variable ($p < 0.05$, highest covariance explained). For each method, this score was obtained by considering CCA between the top 10 functional connections identified using the MAD metric and 10 cognitive scores. Higher-order methods generally display higher correlation scores, except for VA, suggesting a potential advantage. (b) Percentage of covariance explained by the significant latent components ($p < 0.05$) from the CCA analysis. In this case, eFC performed better in three functional networks (DA, VA, and FP), while higher-order approaches excelled in other resting-state functional networks. CCA components were independently assessed for each method. Correlation and covariance estimates were derived by randomly dividing HCP subjects into discovery ($n = 67$) and replication ($n = 33$) sets. Standard errors on the correlation were smaller than 0.01.